# Seamless optical cloud computing across edge-metro network for generative AI

Sizhe Xing[1,2,3,6], Aolong Sun[1,3,6], Chengxi Wang[1,3,6], Yizhi Wang[2], Boyu Dong[1,3], Junhui Hu [1,3], Xuyu Deng[1,3], An Yan[1,3], Yinjun Liu[1,3], Fangchen Hu[4], Zhongya Li[1,3], Ouhan Huang[1,3], Junhao Zhao[1,3], Yingjun Zhou[1,3], Ziwei Li [1,3], Jianyang Shi [1,3], Xi Xiao [5], Richard Penty[2], Qixiang Cheng [2] ✉, Nan Chi [1,3] ✉ & Junwen Zhang [1,3] ✉

The rapid advancement of generative artificial intelligence (AI) in recent years has profoundly reshaped modern lifestyles, necessitating a revolutionary architecture to support the growing demands for computational power. Cloud computing has become the driving force behind this transformation. However, it consumes significant power and faces computation security risks due to the reliance on extensive data centers and servers in the cloud. Reducing power consumption while enhancing computational scale remains persistent challenges in cloud computing. Here, we propose and experimentally demonstrate an optical cloud computing system that can be seamlessly deployed across edge-metro network. By modulating inputs and models into light, a wide range of edge nodes can directly access the optical computing center via the edge-metro network. The experimental validations show an energy efficiency of 118.6 mW/TOPS (tera operations per second), reducing energy consumption by two orders of magnitude compared to traditional electronic-based cloud computing solutions. Furthermore, it is experimentally validated that this architecture can perform various complex generative AI models through parallel computing to achieve image generation tasks.

Recent advancements in generative artificial intelligence (AI) have spotlighted its remarkable capabilities in tackling complex tasks[1,2] such as advanced computer vision[3,4], natural language processing[5,6], and the generation of multimodal content[7]. The backbone of these neural networks' capabilities heavily relies on extensive cloud computational resources[8–11], underpinned by numerous processing units such as graphic processing units (GPUs)[12]. To meet the ever-increasing demand driven by generative AI, significant computational power and storage capacity are called for. Moreover, this growth is at the expense of a substantial energy consumption, with generative AI

reported to consume $9.5×10^{12}$ Wh of ectricity[13] in 2022. According to projections by the international energy agency, data centers could consume a total of $1×10^{15}$ Wh annually by 2026[14]. As the market for generative AI continues to expand, it necessitates reducing operational costs and increasing computational capacity to accommodate both the high computational demands and cost-sensitive user base. However, the continued advancement of this technology is hindered as electronic technology approaches its physical and technical limits[15–19], making further improvements in the speed and efficiency of electronic computing units increasingly challenging[20,21]. Additionally,

---

[1]School of Information Science and Technology, Fudan University, Shanghai, China. [2]Centre for Photonic Systems, Electrical Engineering Division, Department of Engineering, University of Cambridge, Cambridge CB3 0FA, UK. [3]Key Laboratory for Information Science of Electromagnetic Waves (MoE), Fudan University, Shanghai, China. [4]Zhangjiang Laboratory, Shanghai, China. [5]National Information Optoelectronics Innovation Center, Wuhan 430074, China. [6]These authors contributed equally: Sizhe Xing, Aolong Sun, Chengxi Wang. ✉e-mail: qc223@cam.ac.uk; nanchi@fudan.edu.cn; junwenzhang@fudan.edu.cn

computation security and privacy continue to be major concerns[9,22], given that much of the models involved may contain sensitive or confidential information.

To tackle the power and speed limitations of traditional electronic processing units, generative AI and other complex tasks have sparked wide interest in optical computing[20,21,23–25] as a promising solution. This advantage stems from the unique capability of optical systems where forward propagation of computation tasks and light occurs concurrently, embodying the concept of 'propagation' across both computational and physical forms. Recent advancements in optical processing units (OPUs) have showcased their capability to perform a wide range of high-speed, energy-efficient computing tasks, such as data processing[26–28] across complex tasks such as signal processing[29,30], neural network[31–34], and mathematical operations[35–37]. These works primarily focus on localized computation within single optical computing chips, striving to increase computational capacity[28,38,39] on a single chip. However, few studies have leveraged the low-loss and low-latency properties of light[40,41] transmission in the optical computing system, which permit the deployment of optical computing units remotely from edge nodes.

This illustrates a new paradigm in the development of cloud computing: optical cloud computing. Equipped with OPUs instead of GPUs, the optical cloud computing system offers a promising solution to address the challenges associated with storage space, security, and energy consumption that are prevalent in conventional electrical cloud computing systems. Building on previous research[31] demonstrating the feasibility of separating optical computing nodes from model storage nodes, optical cloud computing is expected to conserve cloud storage space while ensuring computing security. Considering the commercial value of the model, optical cloud computing can ensure the privacy of weights. As the weights exist only locally on the edge, the model becomes physically inaccessible to others when the edge nodes go offline, thus eliminating any possibility of unauthorized access. Simultaneously, the energy-saving characteristics of optical computing can significantly reduce the energy consumption of computing centers, thereby reducing the operational costs of cloud computing and further facilitating the development of generative AI. Due to the advancement of optical communication systems, optical network architectures have become widely deployed in metropolitan areas, which enables the optical computing center to provide services to various edge nodes. With the advent of optical cloud computing, the optical network possess the potential to simultaneously support both the communication and computational requirements of various edge nodes across the network in the future.

Here, we propose and demonstrate a seamless optical cloud computing system across edge-metro network by deploying optical computing nodes in the cloud, enabling direct access to OPUs through the existing optical network infrastructure from the edge, as depicted in Fig. 1. The edge-metro network spans metropolitan and edge area, facilitating connections between various edge nodes and the cloud. Enabled by a frequency comb, Mach-Zehnder modulators (MZMs), arrayed waveguide gratings (AWGR), photodetectors (PD) and microrings, the system facilitates task assignment by aggregated edge node from clients and parallel processing by OPUs, while data and weights are simultaneously carried by the frequency tones and transmitted bidirectionally between the optical computing center and edge node. This architecture integrates communication and cloud computing, efficiently utilizing existing network resources to deliver secure optical cloud computing services. By simultaneously transmitting both the model and data via light from the edge to the cloud, this approach not only reduces computational overhead by eliminating the need for deploying additional light sources and data storage, but also significantly enhances computation security. Additionally, since all the required components are commercially available, they provide assurance for the overall stability of the system. It is experimentally validated that each OPU can process data at a rate of up to 3.6 TOPS (tera operations per second), with an estimated power consumption of only 118.6 mW/TOPS. By employing this method, we have implemented handwritten digit recognition and image generation tasks. Furthermore, experimental results indicate that the system can maintain a computational accuracy of 7 bits at an operational rate of 10 GHz.

## Results
### Operating principle of the OPU

Figure 2 illustrates the principle and the structure of the OPU, which consists of the MZMs, microrings, AWGR and PDs. It is based on the cyclic wavelength routing properties of the AWGR[42–45], where the transparent wavelength between the input port $p$ and output port $q$ obey the rule that

$$\lambda_{p,q}(n) = \lambda_0 + (p - q) \cdot \Delta\lambda + n \cdot \Delta\lambda_{FSR} \tag{1}$$

Here, $p$ and $q$ represent the input and output port numbers, respectively. For an AWGR, the total number of input and output ports is always the same. The terms $\lambda_O$, $\Delta\lambda$, and $\Delta\lambda_{FSR}$ denote the center wavelength, the channel spacing, and the Free spectral range (FSR) of the AWGR, respectively. Specifically for the AWGR, $\Delta\lambda_{FSR}$ is the product of the number of ports and the channel spacing $\Delta\lambda$. For detailed derivations, please refer to Supplementary Note 1. Simplifying the filtering characteristics of the AWGR to an impulse function (with an extremely narrow input linewidth), the total optical intensity at the output ports can be expressed as:

$$I_q = \sum_p \sum_n \int_\lambda I_p(\lambda) \cdot \delta\left(\lambda - \lambda_{p,q}(n)\right) d\lambda \tag{2}$$

Here, $I_p(\lambda)$ is the intensity of wavelength in the input port $p$, which can be decomposed into $x(p) \cdot I_O(\lambda)$. $x$ is the signal to be convolved. $I_O(\lambda)$ represents the relationship between the comb tooth intensity and the wavelength for an optical frequency comb with a spacing of $\Delta\lambda$ and a center wavelength of $\lambda_O$. The comb teeth at wavelength $\lambda_{p,q}(n)$ are loaded with the intensity signal $\omega_n(p - q)$, where $\omega_n$ represents the convolution kernel. Therefore, the relationship between the output optical intensity and the wavelength can be simplified to:

$$I_q = \sum_p \sum_n x(p) \cdot \omega_n(p - q) \tag{3}$$

When considering a single FSR, this represents the output of the convolution operation between $x$ and $\omega_n$. When taking into account two FSRs, denoted as 1 and 0, where $\omega_1$ and $\omega_0$ represent the positive and negative parts of the convolution kernel $\omega$, respectively, the signals from the two FSRs are separated by microring filters and then subtracted using a Balanced Photodetector (BPD) to obtain:

$$E_q = \sum_p x(p) \cdot \left(\omega_1(p - q) - \omega_0(p - q)\right) = \sum_p x(p) \cdot \omega(p - q) \tag{4}$$

How the frequencies are processed and the corresponding structures are illustrated in Fig. 2a, b. Figure 2a depicts the step-by-step process of performing optical convolution, where the weights ($w1$, $-w2$, and $w3$) are loaded onto the comb via the waveshaper. The light are then modulated using MZMs. The input light is equally divided into four parts and modulated with input data ($x1, x2, x3, x4$) by four MZMs. At this stage, each x is multiplied by the corresponding weights, resulting in a new vector. The modulated signals are routed through the AWGR, where they are directed to specific output channels based on their wavelengths. Next, at the output end, the signals from two wavelength groups spaced by one FSR are separated by microrings and detected by the balanced photodetector (BPD). Figure 2b shows the device architecture required to implement this operation in our experiment. The left side illustrates the scheme for loading the

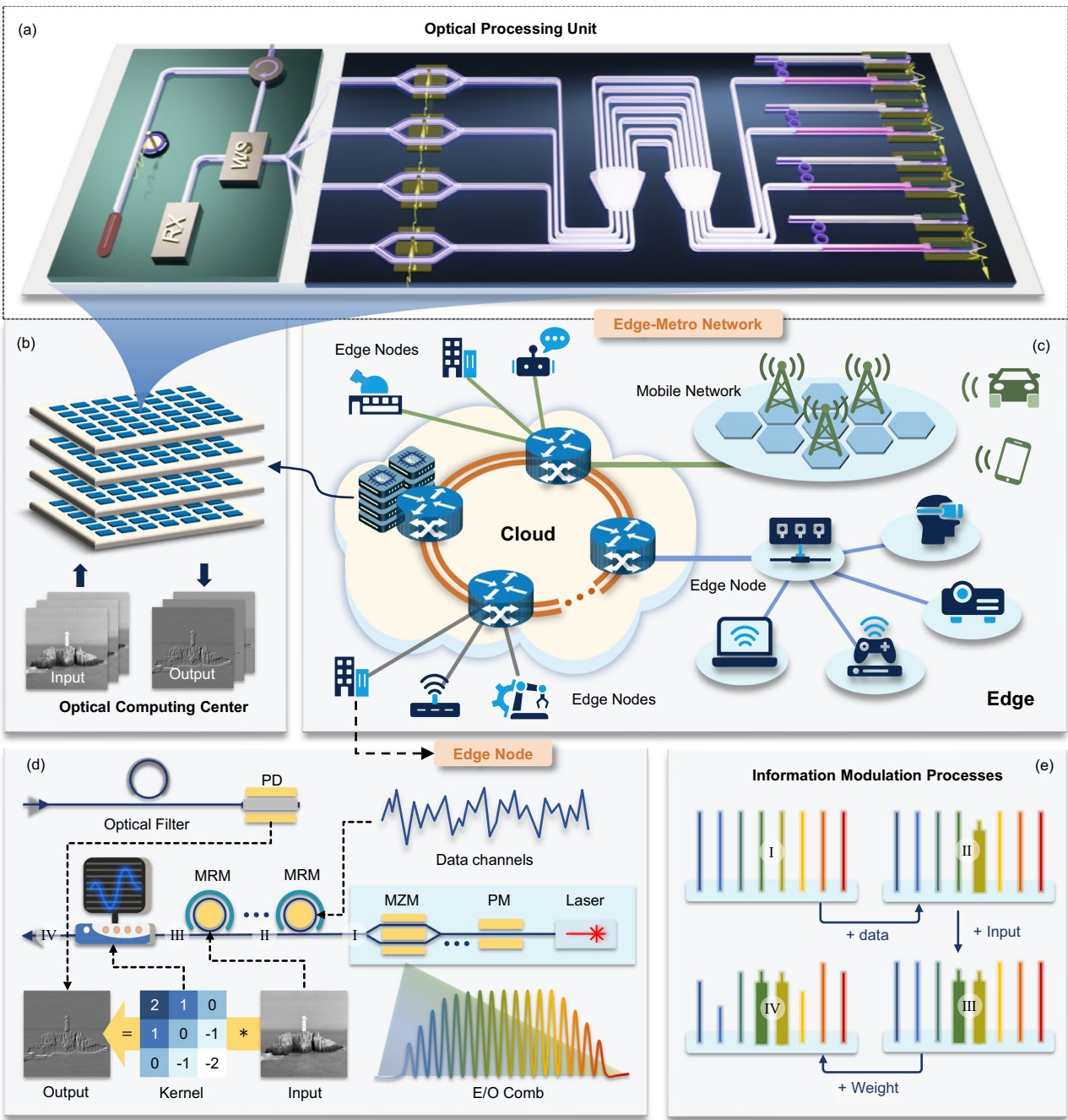

**Fig. 1 | Optical cloud computing architecture. a** Structural layout of an optical computing unit within the cloud-based optical computing center. Comprising an AWGR, MZM array, microring array, and PD array, it performs high-speed optical convolution operations and can transmits the results back to the user. **b** The cloud-based optical computing center is composed of numerous optical computing nodes, enabling parallel processing of extensive image data. **c** The cloud contains optical switching nodes and optical computing centers capable of handling a high volume of tasks from edge nodes and supporting diverse requirements, including computer vision, natural language processing, and multimodal content generation, which require massive convolution operations. **d** The smart optical transceiver deployed at the edge node end supports both access to the optical computing unit and optical communication services. Input, weight, and data are multiplexed onto the optical frequency comb via wavelength division multiplexing. The final output received through a PD. e, Information loading process: data, input, and weight are loaded sequentially onto different frequencies.

weights, where a waveshaper is used to load the weights onto the optical frequency lines.

## Proposed optical cloud computing method

The development of cloud computing and the increasing scale of deep neural networks exhibit a synchronous upward trend[10]. As the computational demands of neural networks exceed the capabilities of home computing units, offloading computing tasks to the cloud has become an inevitable solution. This study leverages the computational properties of light to achieve an optical cloud computing system. This architecture is designed to be deployed within general communication networks, with the computing center positioned in the cloud. The computing data is modulated onto light at the edge node and directly processed by the OPU in the cloud. Since the computing data is only stored at the edge node, computing security is ensured at the physical layer. Leveraging wavelength division multiplexing technology and

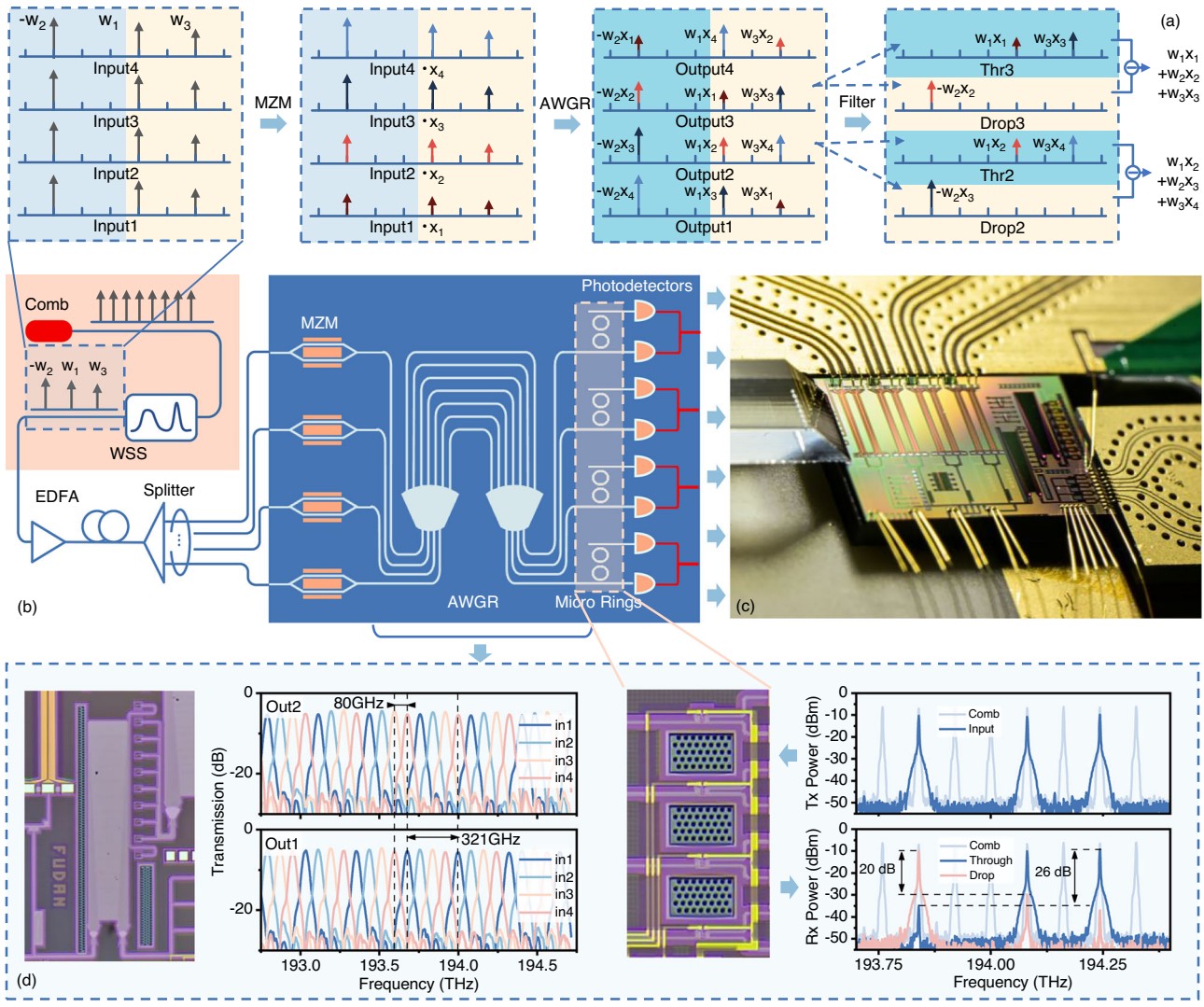

**Fig. 2 | Data loading method and calculation principles of the OPU. a** The principle of convolution process with both positive and negative aspects by utilizing the unique the cyclic wavelength routing properties of an AWGR. It shows how the wavelengths changed in the whole processes. **b** Schematic diagram of the OPU. The orange area represents the weight loading part. The devices used for optical convolution are shown in blue area, including MZMs, an AWGR, microrings, and PDs. **c** Single OPU module packaged on-chip. **d** Micrograph of the AWGR and microring, along with the wavelength transmission graph of two adjacent output ports of the AWGR and the transmission spectrum of the microring.

optical frequency combs, transceivers at edge nodes can be updated to support optical cloud computing while maintaining communication functions. This capability is enabled by the AWGR-based OPU deployed in the cloud, which supports remote weight loading—a valuable feature that sets it apart from many other optical computing architectures. The user transmits both data and computational tasks through an intelligent transceiver, and upon receiving the tasks, the cloud-based optical computing center distributes them across multiple OPUs for simultaneous processing. Since the computing nodes directly use the light transmitted from the users for calculations, this significantly reduces the power consumption associated with lasers. This reduction facilitates the dense deployment of numerous chips, potentially enhancing the computational capacity of the computing centers.

Figure 3 shows how the architecture achieves the image transfer task across the edge node and optical computing center with multiple OPUs. The figure exemplifies a generative residual convolutional neural network, which consists of 15 convolutional layers, including 9 residual blocks. The first step of the approach is to send the inputs and weights from edge to the optical computing center. The input signals are loaded onto the comb through an array of MRMs. And the weights

are loaded through the waveshaper. In one frequency cycle, the system employs specific frequencies to transmit the signal, while others load the weights. Within each frequency cycle, the system employs specific frequencies to transmit the signal, while others load the weights. Concurrent computation tasks are mapped to separate wavelengths, enabling parallelism through spectral resource partitioning. As illustrated in the figure, different convolutions within the same network layer are assigned to independent OPUs for parallel execution. The detailed parallelization methodology, including wavelength assignment protocols and OPU synchronization mechanisms, is comprehensively described in Supplementary Note 4. Convolutional neural network operations are then performed at the OPU, where a waveshaper distinctly separates signals and weights. Signals are directly captured by a PD, and weights are fed into the optical computing module. The final step involves returning the convolutional layer's output to the edge, with the encoded output signals loaded via MRMs. This structure allows multiple OPUs work together to support one complex task.

In this research, an optical frequency comb with a spacing of 21 GHz was utilized, and its spacing was adjusted to 84 GHz using a

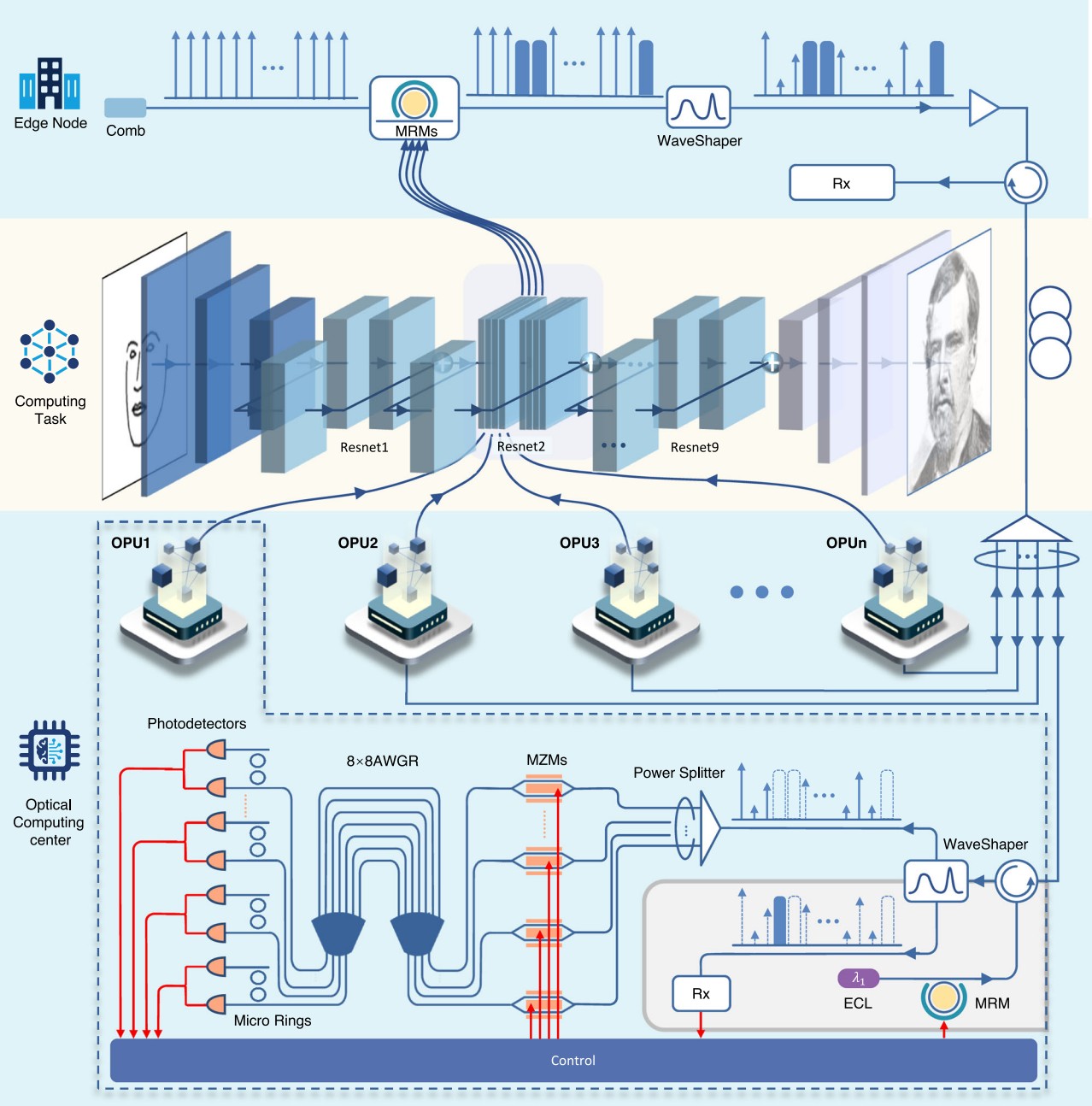

**Fig. 3 | System architecture of optical cloud computing supporting generative AI.** The system comprises two primary components: edge nodes and the optical cloud computing center with multiple OPUs. The transceiver is upgraded to handle both communication and optical computing tasks, consisting of an optical frequency comb laser source, an array of Microring Modulators, a waveshaper, an optical amplifier, and a receiver, which collectively enable upstream data, input and weight loading onto light and reception of downstream signals. Each OPU selectively receives required wavelengths and performs optical convolution computations, structured with two functional sections: computation and communication. The computation section contains a power splitter, MZM array, AWGR, microring array, and PD array, while the communication section includes a receiver and a MRM-based transmitter.

waveshaper to align with the channel spacing of an 8-input, 8-output AWGR. We defined two FSRs of the AWGR as a single cycle for managing weights. In this setup, specific comb teeth were allocated for modulating signals, while others were used for loading weights. For instance, with weight length of 3, the first and last three channels of each FSR were tasked with managing two distinct sets of weights. Positive weights are loaded in one FSR in this cycle, and negative weights in another. The channels not involved in weight loading were used for transmitting signals. When the scale of convolutional operations exceeds the processing capacity of a single OPU, the computation can be partitioned into smaller sub-convolutions and processed in parallel by multiple OPUs. For example, an image convolution task may be divided into multiple one-dimensional convolutions, each assigned to a dedicated OPU for simultaneous execution. A method to decompose image convolution into one-dimensional convolutions is proposed as shown in Supplementary Note 4, allowing the image convolution operation to be distributed across three OPUs for simultaneous computation.

## Integration of optical computing and communication
As a cloud computing system to be deployed across edge-metro networks, it integrates both communication and cloud computing

functionalities. To evaluate the system's performance, both the communication capability and the computing performance of the OPU were tested. Figure 4b displays a frequency comb with an 84 GHz spacing, where regions I and II are used for loading weights, and region III is used for loading input and signals. As experiment setup shown in Supplementary Note 7, a total of 120 images were transmitted, which was transmitted over an 80 km fiber at a rate of 50 Gbps. The communication performance is shown in Fig. 4c, where the maximum received optical power was −15dB, with the corresponding eye diagram shown in Fig. 4d. The maximum supportable power budget was 6 dB, as shown in inset (I), which is still allowed for lossless transmission. When the attenuation reached 7 dB, the signal-to-noise ratio was insufficient to support image transmission, as shown in inset (II).

During the optical convolution process in the OPU, the weights are first multiplied with the signal loaded by the MZM, followed by the summation of results at different wavelengths detected by the PD. The signals separated by the microring are then subtracted. This process encompasses four basic operations: multiplication, addition, subtraction, and multiply-accumulate (MAC). The computational accuracy of these operations at various baud rates was experimentally validated, as shown in Fig. 4e. The precision curves for these operations follow a similar trend, achieving approximately 7 bits of accuracy at a baud rate of 10-Gbaud. As the baud rate increases to 50-Gbaud, the precision decreases, maintaining only about 5.5 bits. At a transmission speed of 10-Gbaud, the error distribution for addition operations under different weights is shown in Fig. 4f. In the experiment, two sets of data, each with eight different levels, were added together, and the theoretical values were compared with the experimental results. The data within the 25–75% range of the experimental results are concentrated in a very narrow interval, where we obtain a standard deviation of 0.0984 in Gaussian error distribution from 4096 addition operations. At this point, the bit precision is 7.12 bits. The successful validation of computational accuracy in four different modes helps determine the range of computational applications possible with this device. This is further confirmed in Fig. 5. At high speeds such as 50-Gbaud, a precision of 5.5 bits can preliminarily support simple applications, such as handwriting digits recognition. However, applications involving more complex models require higher computational accuracy, often needing a precision of 7 bits to be feasible. Given that this optical computing node employs a parallel convolution computation model, it adapts more effectively to higher computational rates compared to methods that use delays in convolution, offering the flexibility to adjust computational precision according to the task. The results of image convolution were further verified under ten different convolution kernels with the Baud rate of 10-Gbuad, in Fig. 4g.

### Experimentally demonstration of image-generation tasks
Generative AI, spearheaded by models like Chatgpt[46], has garnered significant attention recently, leading to numerous companies offering cloud-based model services. These models, due to their extensive computational power demands, cannot be deployed locally and require cloud computing support. High-speed, energy-efficiency and secure optical cloud computing is poised to be a future solution to these challenges. Our experiments have validated the accuracy and feasibility of this approach.

The impact of the limited precision of the OPU on the accuracy of MNIST handwritten digital image classification is investigated in Fig. 5a. To verify the experimental performance of processing all convolutional layers optically, the network was simplified to include only one convolutional layer, as shown in the Supplementary Notes Fig. S16. When the bit precision is less than 5, the accuracy of the network dramatically improves as precision increases, rising from 11% to 90%. Once the bit precision exceeds 5 bits, the accuracy stabilizes around 92%. These trends are confirmed by experimental results in which 100 images from the MNIST dataset are processed through the

optical cloud computing system. In the experiment, the convolution layer was executed optically, while the remaining operations were conducted on a computer, maintaining the same bit precision as the photonic devices. An accuracy of 88% was achieved, as confusion matrix shown in Fig. 5b. This demonstrates that a single OPU can support handwritten digit recognition of An accuracy of 88% with computation speeds[42,43] reaching $6 \times 6 \times 2 \times 50\ GHz = 3.6\ TOPS$.

Complex tasks often require more complex network structures. For generative AI network, tasks need to be parallelized and segmented, as shown in Fig. 5c. Numerous OPUs could simultaneously process computational tasks from the edge node and send the results back, with each OPU handling a specific convolution task within a layer of the network. In the experiment, a convolutional neural network was trained to handle various tasks including season transfer (winter to summer; summer to winter) and semantic segmentation, as shown in Fig. 5f. Additionally, three other tasks are demonstrated in Supplementary Note 8: object generation, mapping aerial photos, and depth estimation. Subsequently, OPUs were used to validate the processing results for these tasks. During the experiment, one OPU handled the computations for the network's first layer, while the computations for other layers were conducted on a computer according to the precision capabilities of the photonic devices. More detailed results are provided in Supplementary Note 8 to further validate the rationale of this approach. With the first layer comprising 64 nodes, the resulting waveforms produced 64 feature convolutional matrix outputs, each containing the extracted hierarchical features of the input image. The output from the first node, corresponding to the red channel of a map image, is illustrated in Fig. 5d. The comparison between the predicted and the measured waveforms reveals a root mean square error of 0.0304. To fully validate the computing performance, images were processed through the first computational layer. This layer consists of 64 nodes, each performing 64 convolutions across the three color dimensions of the images (for grayscale images, only one dimension is involved). The performance comparison between the experiment and simulation results is presented in Fig. 5e, demonstrating that the optical computing system achieves performance comparable to 7-bit precision electronic computing. Also, a selection of the generative AI results is displayed in Fig. 5f, with additional results available in the Supplementary Note 8. In this experiment, to ensure data precision, calculations were conducted at a rate of 10-Gbaud, with each OPU achieving a computational speed of $6 \times 6 \times 6 \times 10\ GHz = 0.72 TOPS$.

Due to experimental limitations, only the calculations for the first convolutional layer were conducted in the optical domain during the experiment, while all other operations were performed on a computer. Although only some operations were carried out experimentally, each OPU in optical computing center is indeed only responsible for a portion of the computational tasks within the network. The strict adherence to the same computational precision for the other layers allows this setup to accurately reflect the practical application scenarios of an optical computing center.

## Discussion
### Computing precision analysis
The precision of the processing unit remains a critical parameter, regardless of the processing speed. Sufficient precision is essential for neural networks to function effectively. These results in optical cloud computing exhibiting lower performance compared to electronic computing, particularly in terms of accuracy, generated image quality, and stability. For relatively simple tasks such as handwritten digit recognition, the accuracy of optical cloud computing reaches 88%, which is lower than the 92% achieved by electronic cloud computing. For image generation tasks, compared to full-precision computing, optical cloud computing achieves SSIM, FID, and LPIPS values of 0.92, 39.8, and 0.23, respectively. These results indicate that while optical cloud computing demonstrates a reasonable level of similarity to state-

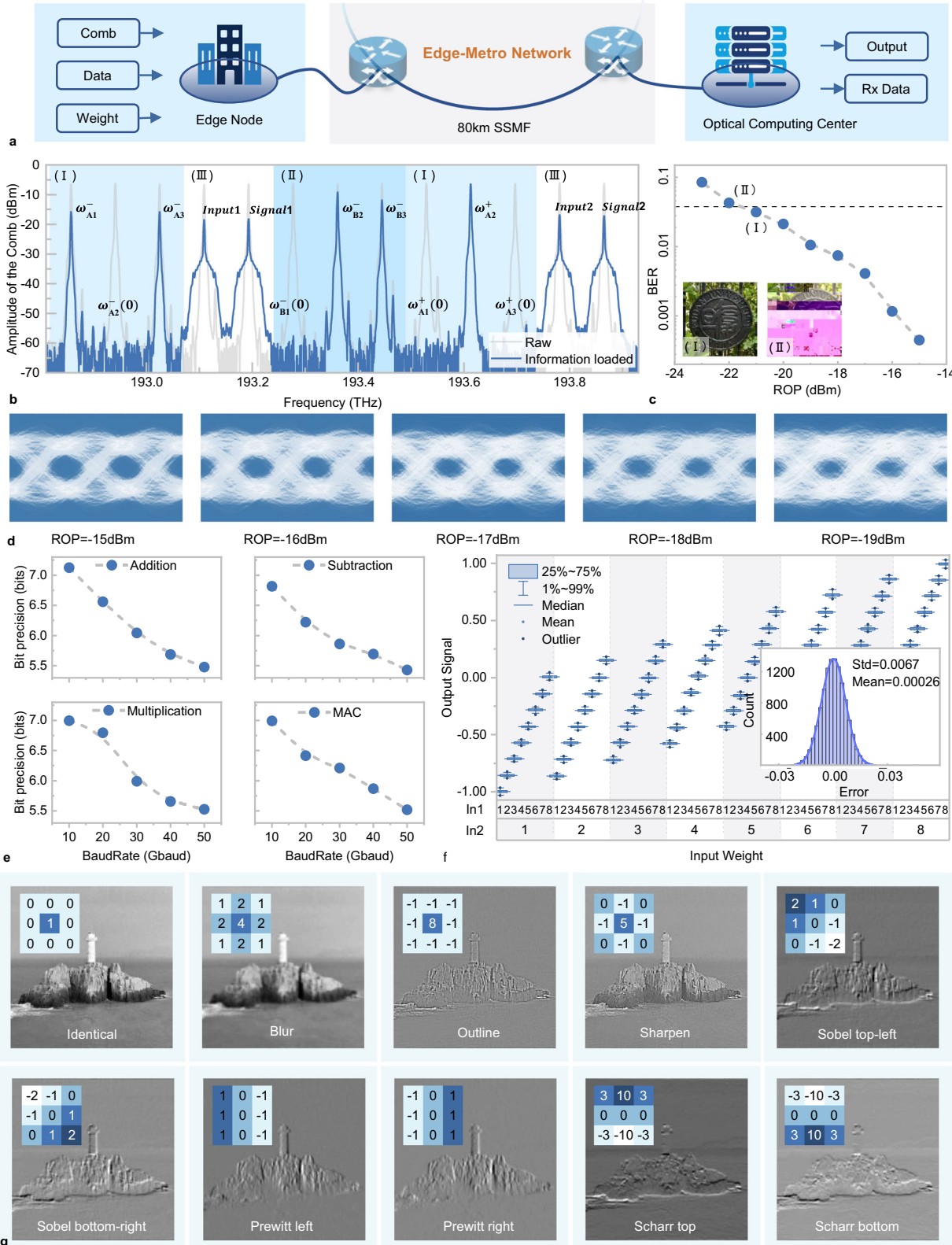

**Fig. 4 | Experimental results of the communication and optical computing.**
**a** Schematic diagram from a single edge node to the cloud computing center. **b** The
frequency comb with weight and signal loaded. **c** The curve of the bit error rate
(BER) as a function of received optical power (ROP) is presented. The image
received under the maximum power budget is depicted in Inset I. When the
attenuation increases further, the incorrectly received image is shown in Inset xII.
**d** The eye diagrams with different ROPs from -15 dBm to −19-dBm. **e** The accuracy
curves of four basic operations as a function of baud rate. **f** The error distribution
for addition operations at a baud rate of 10-GHz. **g** Ten convolution kernels and the
images after convolution.

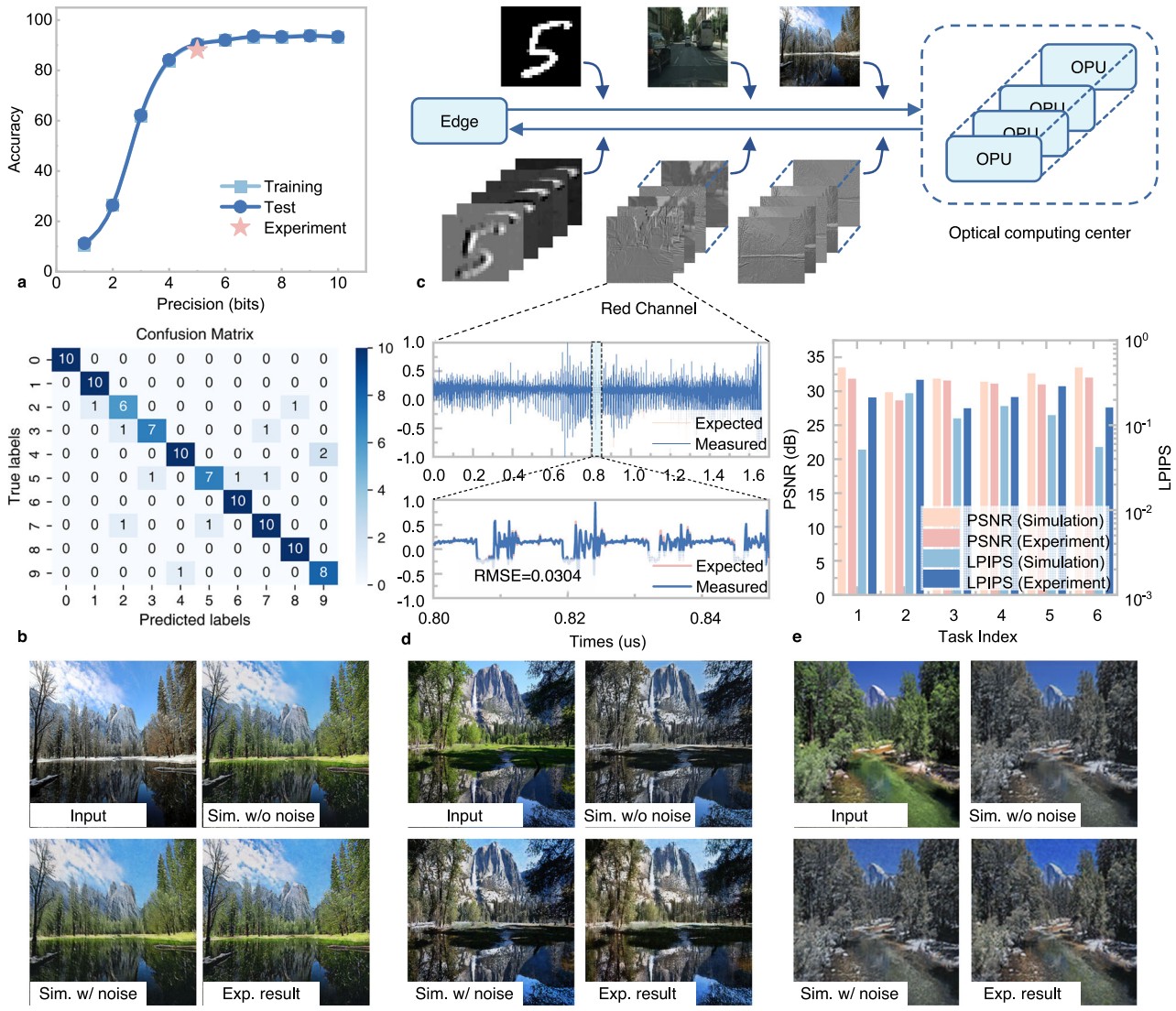

**Fig. 5 | Experimental result of classification and generative AI tasks. a** The accuracy of MNIST handwritten digital image classification with different precision. **b** Confusion matrix for MNIST handwritten digital image classification. **c** Architecture of optical cloud computing system adapted for various tasks. **d** Convolved waveforms from the first layer of the map edge detection task, with the red and blue lines representing the ideal and experimentally generated waveforms, respectively. **e** Performance comparison between simulation and experiment with 6 different generative AI tasks (edges2handbags, edges2portrait, edges2shoes, map2edges, pix2pix-depth and segmentation). **f** Image generation results for season transfer task (winter to summer; summer to winter).

of-the-art electronic computing, its performance is approximately equivalent to that of electronic computing at 7-bit precision. Additionally, the variance of the SSIM parameter for the seasonal transfer task was calculated to be $5.1 \times 10^{-4}$, which is higher than the $3.4 \times 10^{-4}$ observed in 7-bit precision electronic computing. This difference can be attributed to factors such as channel fluctuations in the optical system.

The performance of the optical cloud computing observed in this study is constrained by various factors inherent to the experimental setup, each of which impacts the precision of the final outcomes. Key among these are the resolutions of the primary electrical devices used to generate and receive electrical signals: the arbitrary waveform generator (AWG, Keysight 8194 A with 8 bits) and the oscilloscope (UXR0134A with 12 bits). The lack of an available AWG that offer both high speed and precision represents a significant limitation in the experiment. This ultimately limits our achievable precision to within 8-bit. If a higher-performance AWG, such as one supporting 12-bit precision, is used, the computational accuracy can be further improved. Additionally, the overall bandwidth limitation

of the system, primarily attributed to the modulator and PDs, plays a crucial role. The relationship between precision and baud rate is depicted in Fig. 4c, while the frequency response of the entire computing system is elaborated upon in Supplementary Note 8. Specifically, four Mach-Zehnder Modulator (MZM, T.MXH1.5) and a 100 G photodetector (XPDV4121R-WF-FP) were utilized in this experiment, with the bandwidth of the MZM identified as the predominant factor affecting precision. The packaging process also affects precision by influencing the overall system bandwidth. This is closely related to the spacing and arrangement of high-frequency interconnections. The performance of the packaged device is presented in Supplementary Note 8. Moreover, the nonlinearity inherent in the electrical-to-optical and optical-to-electrical conversions significantly influences the accuracy of computations. This nonlinearity is an unavoidable characteristic of the devices involved. To ameliorate this issue and enhance precision, compensations have been implemented within the system, effectively mitigating the impact of nonlinearity on precision, as demonstrated in Supplementary Note 8.

## Computing speed

Enhancing the computing speed of the processing unit represents a pivotal challenge in deploying generative AI. This challenge is influenced by two principal factors: the frequency of computation and the number of computations per slot. Compared to electronic computing chips, optical computing units have been proven to perform calculations more rapidly and efficiently. In this study, the computation frequency reaches up to 50 GHz, significantly surpassing that of traditional electronic computing units. The bandwidth of optical devices continues to improve. On-chip modulators have already achieved rates exceeding 110 GHz[47], while photodetectors are capable of supporting signals up to 180 GHz[48]. Owing to the utilization of high-bandwidth devices, optical computing benefits from inherently higher frequency, which has the potential to exceed 100GHz-double the current computation frequency. The number of computations per slot is dictated by the inherent characteristics of the computing devices. And the AWGR-based OPU can execute the entire convolution computation within a single clock slot, significantly enhancing the computation speed per unit compared to other convolution units. As AWGR-based OPUs utilize both wavelength and port number for parallel convolutional computations, the peak computation speed can increase quadratically with the number of AWGR ports. However, computing performance will inevitably decline as the scale of the device increases. Considering the accumulation of noise, the SNR of the output signal is primarily influenced by the number of wavelengths received. Therefore, using the same number of wavelengths can be considered to support the computational functions implemented in this paper. With the doubling of computation frequency and convolution kernels of the same size 3, a 64-ports AWGR can reach a peak computing rate of $62 \times 6 \times 2 \times 2 \times 100\,GHz = 148.8\,TOPS$, which is more than 40 times the 3.6TOPS achieved in this experiment.

## Power efficiency

High power consumption in optical computing centers often hinders their large-scale expansion. Various efforts to mitigate this include specific cooling liquids or situating data centers in lakes. Although advanced fabrication technology is enhancing the energy efficiency of electronic chips, they still consume substantial energy—for instance, Nvidia's H200 chip operates at 20.6 W/TOPS[49]. By contrast, considering only the power consumption of the computing units, an AWGR-based OPU exhibits a power efficiency of just 118.6 mW/TOPS, which is significantly lower than that of electronic chips. Since the light sources are positioned at the user end, OPUs in the optical computing center consume even less power. It is noteworthy that as the computational scale of OPUs increases, power consumption scales linearly with component size, while the maximum computational rate scales quadratically[42,43]. Therefore, computational efficiency can be further enhanced with increased component scale, which is further discussed in the Supplementary Note 3. This architecture supports the integration of denser computing chips, paving the way for potentially more compact, large-scale optical computing centers in the future.

The work in this paper centers around a seamless optical cloud computing system across the edge-metro network, where the user's confidential data is modulated into light, then transmitted and directly processed by remote optical computing nodes. This structure allows deployment of optical AI clusters in the cloud, with intelligent transceivers at edge nodes. A single OPU can support a computational rate of 3.6 TOPS with an energy efficiency of 118.6 mW/TOPs, significantly lower than that of electronic computing chips. By employing the delocalized computing scheme, we have implemented multiple different generative AI tasks. Furthermore, this computational capacity can be further enhanced by scaling up the individual OPUs, for instance, by utilizing larger AWGRs. Moreover, the distributed nodes are not limited to AWGR-based computing method. Any approach that supports remote computing can be integrated into the distributed

computing architecture, such as the work based on MZM arrays[31], enabling it to perform various matrix and convolutional operations.

# Methods

## Comb generation and control

In this experiment, a Continuous Wave (CW) wavelength-tunable laser (TSP-400-E0018) with an output power of 16 dBm is utilized to generate the optical comb. The laser's center frequency is tuned to align with the center frequency of the AWGR. Subsequently, the light undergoes modulation using two phase modulators (PM-DV5-40-PFA-PFA-LV) and one intensity modulator (MXAN-LN-40), which are driven by a 21 GHz RF signal produced by a low-noise RF synthesizer (Agilent, 83630B). The RF signals are then amplified to 30 dBm by electrical amplifiers (AT-PA-1840-3330GN) to create a 21 GHz channel spacing comb spanning a bandwidth of 800 GHz (comprising 40 tones with a 5 dB difference between each tone). The power of the comb is further increased to 26 dBm using a gain-flattened EDFA (EDFA, OVLINK, EDFA-C-BA-GF) before being passed through a programmable optical filter (Finisar, WaveShaper 4000S) to achieve a comb with 84 GHz channel spacing, matching the channel spacing of the AWGR. An additional gain-flattened erbium-doped fiber amplifier is then used to elevate the output power of the comb to 24 dBm. A detailed setup of the comb generation is provided in Supplementary Note 7. A 21 GHz RF signal is divided into four paths, with phase control required for three of the signals. After phase matching of the four signals, the RF signal is amplified by an electronic amplifier. All devices used are commercial components, which, compared to integrated optical frequency combs, have the advantage of controllable frequency comb spacing. In the experiment, adjusting the three phase shifters increases the number of output comb lines. The bias voltage of the intensity modulator can control the flatness of the optical frequency comb.

## Details of optical communication experiment

The transmitted images are first compressed to 80 percent of their original size and converted into a series of binary data. This data is then encoded using a Forward Error Correction (FEC) encoder with a code rate of 3/4 and mapped onto PAM signal. The signals are shaped using a Root Raised-Cosine filter and up-sampled to match the sample rate of the AWG (Keysight M8194A with a 120 G sample rate) before being loaded. Following this, the signals are amplified by a RF amplifier and modulated onto a specific frequency of the comb using a fabricated MRM with a bandwidth of up to 110GHz[50,51]. After traveling through 80 km of single-mode fiber, the modulated frequency is filtered by a manually tunable filter (XTM-50-scl-u). The signals are then detected by a high-speed 100 GHz photodetector (XPDV412xR) after amplification by an EDFA to 5 dBm. The output from the photodetector is further amplified by an RF amplifier and sampled by a 59-GHz oscilloscope (Keysight UXR0594BP) with a 256-GSa/s sample rate. On the receiver side, the PAM signal is de-mapped back into a bit sequence. Following FEC and figure decoding, the transmitted image is successfully reconstructed and received.

## Details of optical cloud computing experiment

During the generation of the optical frequency comb, weights are loaded via a waveshaper positioned between two EDFAs (EDFA, OVLINK, EDFA-C-BA-GF), as shown in Supplementary Note 8. Due to experimental constraints with only one waveshaper available, the waveshaper in the edge is also used to filter out the three wavelengths necessary for the computation during the experiment, which is the function of the waveshaper in the OPU. The kernel is loaded onto the wavelengths through the waveshaper, with the intensity of the wavelengths representing the strength of the kernel. Before loading the kernel, the output power of all wavelengths from the optical frequency comb is recorded and mapped into an attenuation lookup table. This attenuation lookup table is then reloaded onto the waveshaper to

equalize the intensity of all wavelengths. Subsequent kernel loading is based on this standardized reference. After passing through a Polarization Controller (PC), the light is equally split into four parts by a 1:4 splitter. During this computational experiment, only three ports of the AWGR are used. The optical power entering each MZM is 18 dBm, where the figures are loaded. The output signal from each MZM, controlled by PCs to match the polarization entering the AWGR. The signal to be processed is generated by an AWG (Keysight M8194A) and then amplified by electronic amplifiers (SHF-S807C) before being modulated onto light using the array of MZM. At the output of the AWGR, the optical signal is amplified to 5 dBm by an EDFA (Amonics AEDFA-23-B-FA) and then converted back into an electrical signal by a photodiode (Finisar XPDV4121R). Finally, the signal is captured by an oscilloscope (Keysight UXR0594BP).

Since the actual central wavelength and wavelength spacing of the AWGR are typically affected by fabrication accuracy and laboratory temperature variations, calibration is required during the experiment to ensure proper operation. Here are the calibration processes we implemented to enhance computational accuracy during the experiment. The first step involves testing the transmission spectrum of the AWGR. Next, the optical frequency comb is adjusted based on the AWGR's transmission spectrum to ensure that the range of the optical frequency comb matches the AWGR's transmission spectrum. It is important to note that in an actual system, this step should be reversed. The optical frequency comb needs to support multiple computation nodes simultaneously, so its frequency should be fixed, while the AWGR should include a temperature control module to align its transmission peaks with the optical frequency comb's frequency domain. To achieve this capability, specially designed tunable AWGs should be applied. By adjusting the refractive index of the rectangular arrayed waveguides or certain sections of the waveguides, the central wavelength and wavelength spacing of the AWGR can be flexibly tuned.

Following this, the bias points of the MZMs need to be adjusted to minimize the nonlinearity of the loaded transmission signal. The adjustment of the bias points is carried out in two steps. In the experiment, the signal is first modulated at the -3dB point, and then fine-tuned to ensure that the bias and modulation depth of each MZM are the same. The first MZM is used as a reference, and the subsequent MZMs are adjusted to transmit signals opposite to the first MZM until the output signal observed on the oscilloscope is zero. The variables that need adjustment include:

**Bias points of the MZM.** Any misalignment in the bias points will result in different nonlinearities in the signals, preventing complete cancellation. Thus, the bias point of each MZM should be finely tuned to make sure the signal is asymmetrical.

**Vpp of the input signal to the MZM.** This affects the modulation depth of the signal, leading to different amplitudes at the receiving end. Precise control is required to ensure consistent modulation depth across all MZMs. This can be verified by observing and checking if the two output waveforms completely cancel each other.

**Time delay of the input signal to the MZM.** Due to different lengths of RF cables, the electrical signals have varying delays when reaching the MZM. These delays cause signal misalignment and errors in the final computation results. Time delay calibration is necessary; a time delay less than the symbol rate will appear as peaks in the time domain on the oscilloscope, which can be eliminated by adjusting the delay.

After compensating for power loss differences between MZMs, polarization calibration is required to ensure that the optical signal entering the chip has the correct polarization, maximizing the coupled optical signal. To mitigate the impact of the bandwidth limitations of MZMs and PDs on high-speed signals, the next step is to transmit a set of test signals to characterize the transmission properties of the link and perform MZM bandwidth compensation. The compensation is implemented using a pre-distortion method, directly loaded through the AWG.

## The architecture of the large-scale model used in this paper
To explore the potential of the optical computing center into generative AIs, the experiment focuses on two image generation architectures: pix2pix[52] and CycleGAN[53], both serving as convolutional neural network test cases. The structures of pix2pix and CycleGAN are shown in Supplementary Note 8. The pix2pix network has been employed for transfer learning, effectively handling four distinct tasks: object generation from sketches, map edge generation, road image segmentation, and image depth detection. Additionally, we utilized CycleGAN to achieve image-to-image translation, specifically for transforming images between winter and summer scenes.

The network architectures and parameters used in both models are identical with those described in the work by Zhu Jun-Yan et al. [53], with code modified from their open-source implementations. To simulate the quantization precision of the optical chip, Gaussian white noise with a specified bit quantization level is added after each convolutional and normalization layer in the models. The testing process consists of three steps: first, training the noise-injected model offline; second, deploying the first convolutional layer onto the optical chip while completing the remaining computations offline; and third, fine-tuning the model using a portion of the data due to differences between the actual and simulated noise. The final experimental results are obtained after the fine-tuning process.

## Power consumption
The power consumption of the OPU consists of two parts: the power of the transceiver, the power of the computing part and the power of the electrical control devices. They are mainly derived from the tunable optical filter, modulators, lasers, photodetectors, EDFAs and other electrical devices, as detailed in the Supplementary Note 8. To ensure the long-term operation and stability of the system, additional temperature controllers are required to maintain the functionality of the chip. This aspect has been considered in both our experiments and manuscript, with detailed records provided in the Supplementary Information. Here, $P_{laser}$ indicates the emission power of the laser at 16 dBm, while $P_{TEC}$ refers to the power of the thermoelectric cooler, approximately 1.3 mW. The wall-plug efficiency $\eta \approx 0.3$ is defined as the energy conversion efficiency from electrical power to optical power, leading to a calculated power consumption for a single laser of approximately 137.7 mW. The power consumption $P_{MZM}$ of a MZM is calculated from the product of the bias voltage and current. As each MZM operates at a different bias voltage, the average power consumption per MZM is about 5 mW. $E_{MRM}$ represents the power consumption of MRMs, composed of the power from biasing and heaters. With a bias current of 9uA and voltage of 2 V, the power consumption is dominated by the heaters. In the experiment, with a heater voltage of 2 V and a current of 2.9 mA, the power consumption of a single MRM is 5.8 mW. $P_{Pd}$ is the energy consumption of the photodetector, estimated from $P_{pd} = RV_{bias}P_{rec}$, where $R \approx 0.65 A/W$ is the responsivity of the photodetector, $V_{bias} = 2V$, and $P_{rec} = 3mW$. Thus, $P_{Pd}$ is approximately 3.9 mW. EDFAs, placed before the PDs to boost the received optical power, use a broadband light source for pumping, with a central frequency $\lambda_p$ close to the frequency of signal light $\lambda_s$ around 1550 nm. The input optical power $P_{in}$ is 0.1 mW, and the output power $P_{out}$ is 3.1 mW, with an energy conversion efficiency $\eta \approx 0.3$. Integrated multi-functional optical filters based on MZIs have been widely reported, which typically consume about 20 mW and are sufficient for this system. To ensure a fair comparison, the power consumption of the electrical control module is also considered. This is mainly

influenced by the DAC power, which is 40-mW. In the optical computing module, a total of 8 DACs and 6 ADCs are needed, consuming a combined power of 320.12-mW. In the communication module, 1 DAC and 1 ADC are required, occupying 20.04 mW of power. After accounting for the number of each component, the total power consumption is calculated to be 614.36-mW. However, when only considering the power consumption of the computation part, excluding the transceivers, the total energy consumption amounts to 426.92-mW. The energy efficiency can be calculated as $426.92 mW/3.6 TOPS = 118.6 mW/TOPS$. For the long-term operation, the thermal stabilizer for the AWGR is also essential beyond the experimental setup. However, its power consumption is minimal, requiring only 600 $\mu W^{54}$.

## Chip fabrication

All the photonic chips tested in the study are fabricated in Chongqing United Microelectronics Center. It is based on a 200 mm SOI substrate with 2 μm BOX and 220 nm top silicon, with the 450-nm-wide nanophotonic waveguides with a loss of less than 1.5 dB cm−1. The chip packaging was carried out by national optoelectronics innovation center.

## Data availability

Source Data file has been deposited in Figshare under accession code DOI link[55].

## Code availability

The open-source code of the seamless cloud computing system is available at https://github.com/szxing21/SCOC.

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

## Acknowledgements

This work was supported by the National Key Research and Development Program of China (2023YFB2905700), National Natural Science Foundation of China (under Grants 62171137 and 62235005), the Natural Science Foundation of Shanghai under Grant 24ZR1490500, the European Union's Horizon 2020 research and innovation program, project INSPIRE (101017088), and the UK EPSRC through project QUDOS (EP/T028475/1).

## Author contributions

These authors contributed equally to this work: Sizhe Xing, Aolong Sun and Chengxi Wang. S.X. conceived the basic idea. S.X., A.S., Q.C., and J.Zhang contributed to the basic framework and feasible technical route the paper. S.X. and A.S. designed the photonic chip. C.W. designed all the neural networks used in the experiments. Y.W. contributed to the feasibility analysis of the principles. B.D and Y.L. designed and constructed the electro-optic frequency comb. S.X. completed the experimental test code with assistance from J.H. S.X. was responsible for experimental design and conducted the experimental verification of the interconnection section with the assistance of A.Y., Zhongya L., and O.H. S.X. also conducted photonic computing experiments with the help of A.S., X.D. and J.Zhao. Y.W., F.H., Y.Z., Ziwei L., J.S., and X.X. analyzed and guided the adjustment of experiments. The project was carried out under the supervision of J.Zhang, Q.C., R.P., and N.C.. S.X. wrote the manuscript and revised it based on comments from all authors.

## Competing interests

The authors declare no competing interests.
