## [Transparent Peer Review file · Nature Communications]

Seamless Optical Cloud Computing across Edge-Metro Network for Generative AI

Corresponding Author: Professor Junwen Zhang

Version 0:

Reviewer comments:

Reviewer #1

(Remarks to the Author)

This paper presents a novel cloud optical computing system utilizing specifically designed optical processing devices to enable the combination of remote optical computing and optical communication. The proposed architecture addresses the growing demand for computing power in the era of generative AI. By leveraging the advantages of optical computing, the proposed system achieves energy-efficient cloud computing while seamlessly integrating with existing optical networks, such as metro networks.

The paper is well-structured and well-written, with the authors providing detailed results to support their methodology. The idea is interesting and the topic is of significance to cloud-based AI computing networks; however, there are several points of this work that need clarification:

1. AWGRs are typically fabricated with fixed parameters, such as wavelength spacing and central wavelength, which cannot be modified. This poses a general challenge for AWGR-based optical computing components. I suggest the authors to discuss how this issue was resolved in experiments and its constraints for future applications.
2. The scalability of optical computing devices is now a crucial factor. Expanding the computational scale of AWGR-based devices could significantly increase their physical size. Please elaborate on the scalability of this approach.
3. In the supplementary materials, the authors mention the generative AI model and architecture used but do not provide details on parameter sizes or network structures. These details are critical for the reproducibility of the study. It is recommended that the authors include the specific network architecture. Additionally, the precision requirements of generative AI models are crucial in determining whether optical computing can be effectively applied to generative AI. The paper demonstrates that feasible generative AI results can be achieved with 7-bit precision. Can this network architecture achieve similar results under different computational precision levels?
4. In the discussion section, the authors state that "With the doubling of computation frequency...". However, this statement is inaccurate, as increasing the computation frequency could also impact computing accuracy. This assumption needs further detailed analysis.

(Remarks on code availability)

The source file contains two parts of the code. The authors have shared both the code for designing the device and creating its layout, as well as the code for the generative AI model. The code of generative AI includes a README file and can be executed directly. However, the device design code lacks a README file. Upon opening the project, I found that running it requires a license from Luceda, please provide further details on the license.

Reviewer #2

(Remarks to the Author)

Summary:

The authors propose and experimentally demonstrate an optical cloud computing system within an edge-metro network. Their approach involves transmitting input data and model weights from the edge to optical computing nodes in the cloud, performing computations optically, and returning the results to the edge. Since data remains in the optical domain throughout the process, the system significantly reduces the power consumption of lasers. Their experiments demonstrate orders-of-

magnitude lower energy consumption compared to conventional electronic cloud computing.

Strengths:

I appreciate the authors' effort to eliminate optical-electronic conversions and leverage optics for both communication and computation. This is a promising direction for low-power AI acceleration. The experimental methodology is well-documented, and the results provide strong initial validation of the concept. However, I have several concerns and questions regarding scalability, numerical precision, and system feasibility for modern AI workloads, as listed below.

Concerns & Questions:

1. Scalability & Multi-OPU Processing

From what I understand, the experimental results focus on a single convolution operation or a single convolutional layer. However, modern generative AI models rely on deep neural networks with multiple layers and millions to billions of parameters.

The authors suggest that multiple OPUs could be used to process different layers, but this is not experimentally demonstrated. More details on the parallelization strategy would strengthen the paper.

If multiple OPUs are used sequentially, does the optical signal require amplification after each layer? If so, would this introduce power and latency overheads that reduce the system's efficiency?

Most deep learning architectures include non-linear functions (e.g., ReLU, GELU, Softmax) between layers. While there are examples of optical non-linearity in literature, they come with challenges such as signal amplification, integration of multiple materials, and packaging issues. How do the authors plan to integrate non-linear functions? Would this require switching back to electronic computation at certain stages?

If a single layer has more parameters than one OPU can handle, how would intermediate results be stored or aggregated? Would this introduce additional latency or require hybrid optical-electronic processing?

2. Feasibility for Modern AI Models

The MNIST digit classification task is relatively simple and can be performed with very low computational precision.

Additionally, the GAN architectures used (from 2016-17) are significantly smaller and less complex than today's generative AI models (diffusion models, transformers).

How well do these architectures represent real-world AI workloads in edge devices?

Could the proposed system handle state-of-the-art generative AI models?

What modifications would be required to scale this system to modern deep-learning architectures?

3. Numerical Precision & Data Representation

The reported 7-bit integer precision may be sufficient for MNIST classification, but modern generative AI models typically use low-precision floating-point formats (e.g., FP8, FP16) to preserve dynamic range.

Integer representation is known to struggle with dynamic range, especially in generative tasks.

If higher precision is needed, what modifications would be required to support it?

(Remarks on code availability)

Reviewer #3

(Remarks to the Author)

This work proposed and demonstrated a seamless optical cloud computing system across edge-metro network by deploying optical computing nodes in the cloud, enabling direct access to OPUs through the existing optical network infrastructure from the edge. The manuscript is well-written, and the idea is interesting and innovative. Below, I have detailed my concerns and suggestions, which I hope will be useful for the authors to improve this manuscript.

1. As the authors mentioned in the paper, only one convolutional layer was conducted in the optical domain during the experiment, while all other operations were performed on a computer. My concern is that this cannot unlock the full potential of optical cloud computing. Particularly, the optical convolution process by the current method is relatively straightforward. I would expect multilayer convolution in the optical domain. Without this, I cannot recommend this paper to be published in Nature Communications.

2. In this paper, the computational accuracy, speed and energy efficiency of optical cloud computing are described in detail, and it is pointed out that optical cloud computing can achieve a variety of image generation tasks with low energy consumption, but the specific effects of task realization are not compared. Without the specific effects of the image generation task, and the lack of comparative data (such as the accuracy of the generated image, loss of structure, detail retention, etc.), it is difficult to fully evaluate the advantages and disadvantages of optical computing in real-world tasks. It is suggested that comparative experiments of one or more generation tasks can be added in this paper to compare the performance of optical cloud computing and electronic cloud computing in the quality, speed, accuracy and stability of image generation.

3.The paper mentions that the low energy consumption of optical cloud computing centers can significantly reduce the operating costs of cloud computing, thus promoting the development of generative AI. However, the hardware of optical computing systems is often more complex than traditional electronic computing, and the manufacturing process is relatively new, so can its stability be guaranteed in a long time, high-load cloud computing environment? In addition, are optical computing systems likely to require additional cost inputs and special maintenance to ensure their long-term stable operation?

4.In optical cloud computing systems, with the increase in the number of parallel OPUs, time delay calibration and phase calibration can ensure synchronization between OPUs and the consistency of the optical signal phase, but does the calibration process introduce additional delays, especially when the optical path length and signal phase need to be precisely adjusted? If so, do these delays significantly affect the overall performance of the system, especially in high-speed parallel computing and real-time generation tasks?

5.Minor errors on the caption of Fig. 5. “experimental result” should be “Experimental results”.

6.Line 42, “to consumed” should be “to consume”.

7.Line 295, “n the experiment” should be “in the experiment”.

(Remarks on code availability)

Version 1:

Reviewer comments:

Reviewer #1

(Remarks to the Author)

The authors have thoroughly addressed all the reviewers' comments and made detailed revisions throughout the manuscript. They have demonstrated the scalability potential of the proposed system and provided comprehensive data and code to facilitate reproducibility. The issues previously raised regarding the code have also been fully resolved. To sum up, I believe the current version of the paper has reached the level of quality required for publication in Nature Communications. The idea presented in this paper is interesting and the topic is of significance to future cloud-based AI computing networks. I have no further questions.

(Remarks on code availability)

The issues previously raised regarding the code have been fully resolved.

Reviewer #2

(Remarks to the Author)

The authors have addressed all my concerns and questions.

(Remarks on code availability)

Reviewer #3

(Remarks to the Author)

The authors have given comprehensive and clear responses to my previous comments. I have no further comments and recommend the publication of this paper.

(Remarks on code availability)

Response to Referees Letter

Nature Communications (Manuscript ID: NCOMMS-24-86463)

We would like to sincerely thank all the reviewers and the editor for their time, constructive feedback, and insightful comments. Your suggestions have been invaluable in helping us improve the clarity, depth, and overall quality of our manuscript. We have carefully addressed each point raised and revised the paper accordingly. We truly appreciate your efforts and the opportunity to further refine our work.

Before commencing our point-by-point response, we wish to clarify the naming and numbering conventions employed across the different files to ensure clear distinction.

Figures, tables, and references have been designated as follows:

1. Fig. R1 refers to the figures in the response letter;
2. Fig. 1 refers to the figures in the main manuscript;
3. Fig.S1 refers to the figures in the supplementary information;
4. Table R1 refers the tables in the response letter;
5. Table 1 refers to the tables in the main manuscript;
6. Table S1 refers to the tables in the supplementary information;

Reviewer #1

General Comment:

This paper presents a novel cloud optical computing system utilizing specifically designed optical processing devices to enable the combination of remote optical computing and optical communication. The proposed architecture addresses the growing demand for computing power in the era of generative AI. By leveraging the advantages of optical computing, the proposed system achieves energy-efficient cloud computing while seamlessly integrating with existing optical networks, such as metro networks.

The paper is well-structured and well-written, with the authors providing detailed results to support their methodology. The idea is interesting and the topic is of significance to cloud-based AI computing networks; however, there are several points of this work that need clarification.

Response to General Comment:

We sincerely appreciate the reviewer's valuable suggestions on our paper. We have carefully reviewed these comments and believe they significantly contribute to improving our work. In response, we have made detailed revisions based on the reviewer's feedback. The specific responses and modifications are presented below.

Comment 1:

AWGRs are typically fabricated with fixed parameters, such as wavelength spacing and central wavelength, which cannot be modified. This poses a general challenge for AWGR-based optical computing components. I suggest the authors to discuss how this issue was resolved in experiments and its constraints for future applications.

Response to Comment 1:

Thank you very much for the authors' suggestion. This is indeed a crucial point for practical applications, and it greatly helps us enhance the practicality of our solution. The AWGR itself is indeed a relatively fixed device, as its parameters, such as the central wavelength, channel spacing, and routing relationships, are typically determined during the design phase. However, this does not hinder the flexibility of the structure. There are two main reasons for this:

1. In our designed OPU structure, the AWGR serves solely as a wavelength routing device, while the more critical parameter tuning is handled by other components, such as the MZM and micro-ring resonators. These devices are tunable and can be flexibly controlled via external voltages to achieve various operations. For example, in our experiments, the final electrical signal output from the BPD is given by:

$$E_q = \sum_p x(p) \cdot (\omega_1(p - q) - \omega_0(p - q)) = \sum_p x(p) \cdot \omega(p - q)$$

In this equation, $x(p)$ is modulated onto the optical signal by the MZM, while $\omega(p - q)$ is loaded onto specific optical frequency comb teeth by the WS. These decisive parameters are independent of the specific parameters of the AWGR. Therefore, even a completely fixed AWGR does not cause any inconvenience to the experimental results. By aligning the frequencies of the optical frequency comb with the transmission peaks of the AWGR in the experiment, we ensure the optimal performance.

2. The AWGR is not entirely non-adjustable in practice. In our experiment, three key parameters of the AWGR affect the calibration process: free spectral range (FSR), channel spacing, and central wavelength. By changing the refractive index of part of the AWGR, all of them can be adjusted.

The FSR determines the routing periodicity of the AWGR. In our optical computing

unit, this parameter governs the interval between positive and negative cycles, which directly impacts the wavelength range design of the optical frequency comb. According to the AWGR design principle, the FSR can be derived as:

$$\Delta FSR = \frac{n_c \lambda}{k n_g}$$

where λ is the central wavelength, n_c is the waveguide refractive index, k is the diffraction order, and n_g represents group dispersion. This equation indicates that the FSR can be tuned by modifying the waveguide refractive index.

The central wavelength affects the alignment between the light source and the computing unit. While wavelength mismatches can be compensated by adjusting the light source, we have also enhanced flexibility by enabling direct tuning of the computing unit's central wavelength to match the source. As derived in Supplementary Note 1, the central wavelength satisfies:

$$n_c(\lambda_0) \Delta l_m = k \lambda_0$$

where Δl_m is the length difference between arrayed waveguides. Notably, the central wavelength is also influenced by the waveguide refractive index. In Ref. ¹, a graphene-based fast-tunable AWGR device is proposed by dynamically controlling the refractive index of part of the waveguide. The modified relationship becomes:

$$n_c(\lambda_0) (\Delta l_m - \Delta l_G) + n_G(\lambda_0) \Delta l_G = k \lambda_0$$

Here, Δl_G denotes the length difference of the graphene-coated waveguide segments, and n_G represents the tunable refractive index of the graphene-coated regions. This approach enables precise control over the central wavelength. Similar strategies can also be applied to adjust the channel spacing.

Revision to Comment 1:

In the method section, some analysis is added to illustrate the importance of the calibration of the AWGR. A design guidance of the tunable AWGR is also provided in this section:

“Since the actual central wavelength and wavelength spacing of the AWGR are typically affected by fabrication accuracy and laboratory temperature variations, calibration is required during the experiment to ensure proper operation. Here are the

calibration processes we implemented to enhance computational accuracy during the experiment. The first step involves testing the transmission spectrum of the AWGR. Next, the optical frequency comb is adjusted based on the AWGR's transmission spectrum to ensure that the range of the optical frequency comb matches the AWGR's transmission spectrum. It is important to note that in an actual system, this step should be reversed. The optical frequency comb needs to support multiple computation nodes simultaneously, so its frequency should be fixed, while the AWGR should include a temperature control module to align its transmission peaks with the optical frequency comb's frequency domain. To achieve this capability, specially designed tunable AWGs should be applied. By adjusting the refractive index of the rectangular arrayed waveguides or certain sections of the waveguides, the central wavelength and wavelength spacing of the AWGR can be flexibly tuned.”

Comment 2:

The scalability of optical computing devices is now a crucial factor. Expanding the computational scale of AWGR-based devices could significantly increase their physical size. Please elaborate on the scalability of this approach.

Response to Comment 2:

Thank you for the reviewer's suggestion. This is indeed a critical factor affecting the practicality of our approach. Conducting further comparisons can significantly enhance the feasibility of our solution. Scalability has always been one of our key considerations, and we have taken this issue into account in our design. In fact, as the size of the AWGR increases, its physical footprint also expands. We have illustrated this trend in Fig. R1.

Fig. R1 Design size of AWGR for different wavelength spacings and channel numbers. From bottom to top: 4, 8, 16, 32, and 64 channels.

From the figure, it can be observed that as the number of channels increases, the device size also increases. This increase in size is primarily attributed to the elongation of the short side due to the expansion of the star coupler. Even when scaling up to a 64×64 configuration, the length of the short side does not exceed 1.8 mm. However, the increase in channel number does not significantly affect the length of the long side. This is determined by the design principles of the AWGR. In the AWGR design process, the volume of the rectangular waveguide depends on the product of the number of waveguides and the difference in length between them:

$$N\Delta l_m = N_A \frac{n_s d_i \lambda_0}{\Delta \lambda n_g}$$

Under the same medium and wavelength band, n_s , λ_0 , and n_g , remain constant, while the minimum value of d_i depends on the mode field size in the transmission medium. The figure illustrates two AWGR with different channel spacings. As the number of channels increases, both arrays exhibit a similar trend of growth in the short side length, while the long side length remains the same. This comparison demonstrates the universality of this trend.

Table R1 Device size and the peak computational speed of the AWGR versus channel numbers with the channel spacing of 100GHz

Ch. Number	Ch. Spacing	Max Input	Max Kernel	Device Width	Computations / Clock Cycle
4	100 GHz	4	4	368 um	64
8	100 GHz	8	8	464 um	256
16	100 GHz	16	16	650 um	1024
32	100 GHz	32	32	1036 um	4096
64	100 GHz	64	64	1851 um	16384

Table R1 demonstrates the impact of increasing the number of channels on device size and peak computational capacity under the same channel spacing (100 GHz). It can be observed that as the number of channels increases from 4 to 64, the device size increases by 400%, while the computational speed increases by 25500%. Consequently, the computational efficiency per unit area is significantly improved. Although scaling up the AWGR leads to a certain increase in physical size, the substantial enhancement in computational speed makes this trade-off worthwhile.

Fig. R2 Computational speed as a function of device size with a, 50G AWGR; b, 100G AWGR.

Revision to Comment 2:

We have added the contents of Fig. R1 and Table R1 in Supplementary Note 1 as Fig. S2 and table S1, in the revised version of the supplementary information. We have also added the corresponding texts regarding the description of these figures in Supplementary Note 1:

Scalability of the AWGR-based processing unit

The approach of using an AWGR for convolution computations has a limitation in that the size of the device is large and, theoretically, difficult to significantly reduce. By combining Equations (10) and (12), it can be determined that:

$$R = \frac{n_s d_i d_a \Delta \lambda_{FSR}}{2 \lambda \lambda_0} \quad (14)$$

This indicates that increasing the number of channels will significantly enlarge the device size due to the expanded FSR. The specific impacts are illustrated in Fig. S2. Fig. S2a shows the AWGRs with a 50 GHz channel spacing at varying channel numbers, while the Fig. S2b displays the corresponding dimensions for a 100 GHz channel spacing. From the figure, it can be observed that as the channel number increases, the device size grows accordingly. This scaling is primarily driven by the elongation of the short side caused by the expanded star coupler structure. Notably, even in a 64×64 configuration, the short side length remains below 1.8 cm. Table S1 quantifies the impact of channel number scaling on device size and peak computational capacity under a fixed channel spacing of 100 GHz. As the number of channels increases from 4 to 64, the device size scales by 400%, while the computational speed exhibits a remarkable 25,500% enhancement. This results in a significant improvement in computational efficiency per unit area. Although enlarging the AWGR introduces a moderate increase in physical footprint, the substantial gains in computational throughput justify this trade-off, demonstrating the scalability and practical viability of the proposed architecture.

Comment 3:

In the supplementary materials, the authors mention the generative AI model and architecture used but do not provide details on parameter sizes or network structures. These details are critical for the reproducibility of the study. It is recommended that the authors include the specific network architecture. Additionally, the precision requirements of generative AI models are crucial in determining whether optical computing can be effectively applied to generative AI. The paper demonstrates that feasible generative AI results can be achieved with 7-bit precision. Can this network architecture achieve similar results under different computational precision levels?

Response to Comment 3:

We sincerely thank the reviewer for the constructive feedback, which can help to significantly improve the clarity and reproducibility of this work. Regarding the details of the generative AI model architecture in the supplementary materials, we fully agree to add the details of the specific parameter sizes and network structures of the study. In the revised supplementary materials, we have firstly comprehensively documented the network architecture. The Table R2 provides comprehensive details of the network architecture, specifying the layer composition, parameter number, and computational cost for each module. This ensures reproducibility while highlighting the scalability of the proposed design.

Table R2 Network Architecture and Detailed Parameters. This network consists of 1 input/output module, 2 downsampling/upsampling modules, and 8 residual modules.

Module	Network Layers	Parameters	Computational cost(FLOPs)
Input	ReflectionPad2d(1)	/	/
	Conv2d(3,64,3,1,0)	1,792	226.49M
	InstanceNorm2d(64)	/	25.17M
	ReLU()	/	/
DownSampling No. 1	Conv2d(64,128,3,2,1)	73,856	2,415.92M
	InstanceNorm2d(128)	/	12.58M
	ReLU()	/	/

DownSampling No. 2	Conv2d(128,256,3,2,1)	295,168	2,415.92M
	InstanceNorm2d(256)	/	6.29M
	ReLU()	/	/
Resnet Block ×8	ReflectionPad2d(1)	/	/
	Conv2d(256,256,3,1,0)	590,080*8	4,831.84M*8
	InstanceNorm2d(256)	/	6.29M*8
	ReLU()	/	/
	ReflectionPad2d(1)	/	/
	Conv2d(256,256,3,1,0)	590,080*8	4,831.84M*8
	InstanceNorm2d(256)	/	6.29M*8
UpSampling No. 1	ConvTranspose2d(256,128,3,2,1,1)	295,040	9,663,68M
	InstanceNorm2d(128)	/	12.58M
	ReLU()	/	/
UpSampling No. 2	ConvTranspose2d(128,64,3,2,1,1)	73,792	9,663,68M
	InstanceNorm2d(64)	/	25.17M
	ReLU()	/	/
Output	ReflectionPad2d(1)	/	/
	Conv2d(64,3,3,1,0)	1,731	226.49M
	Sigmoid()	/	/
Total	/	10,182,659	102.10G

On the question of computational precision requirements, we appreciate the reviewer’s insight into the importance of evaluating optical computing’s applicability across varying precision levels. While the current results demonstrate feasible generative AI performance at 7-bit precision, we have conducted further experiments to systematically assess the impact of precision. These tests reveal that the architecture maintains acceptable output quality at lower precision for the same tasks.

Fig. R3 The performance of the network's output under varying computational precision levels. In the revised manuscript, we demonstrate how computational precision impact image generation performance. Notably, the primary effect of precision reduction shown as increased noise in generated images rather than the content quality. As exemplified in **Fig. R3**, all outputs successfully fulfill the intended tasks despite various level of noise. This performance stems from the inherent noise compatibility of our architecture. By decomposing a large-scale model into a series of simple convolutional tasks, the proposed architecture effectively mitigates noise accumulation in all-optical networks. Consequently, the limited computational precision inherent to photonic systems does not significantly compromise task performance. At a computational precision of 6 bits, the season transfer task successfully converts grass and trees into their corresponding seasonal appearances, while in the semantic segmentation task, objects such as cars, roads, trees, and buildings are accurately differentiated by distinct colors. However, due to the limited precision, the output exhibits significant noise, particularly visible in the sky, clouds, and large color blocks. When the precision is increased to 8 bits, the reduction in noise leads to a marked improvement in image quality, approaching the level of outputs generated without added noise. These results validate the high noise tolerance of the proposed architecture, a critical enabler for deploying optical cloud computing in generative AI applications.

Revision to Comment 3:

We have added the contents of Table R1 in the Supplementary Note 8. We have also added the corresponding contents in Supplementary Note 8 in page 14:

“The network architecture used in the experiments is detailed in Table S3, which provides comprehensive information on the layer composition, parameter numbers, and computational costs for each module. Additionally, the source code is available in the supplementary materials for further reference and reproducibility.”

Comment 4:

In the discussion section, the authors state that "With the doubling of computation frequency...". However, this statement is inaccurate, as increasing the computation frequency could also impact computing accuracy. This assumption needs further detailed analysis.

Response to Comment 4:

Thank you for the valuable suggestion, which significantly enhances our manuscript. Indeed, while improving computational speed, the precision of calculations tends to decrease. Therefore, achieving the same level of computational performance requires devices with higher bandwidth. Specifically, both the modulator and photodetector in the system need to be upgraded to higher bandwidths. Fortunately, existing modulators^{1,2} and PDs³ are already available to meet this requirement. However, due to limitations in our laboratory equipment, we were unable to implement the same precision design at higher speeds. Nevertheless, recent studies have continuously pushed the boundaries of the clock frequency limits in optical computing. For instance, a compact silicon photonic computing engine achieved a clock rate of 60 GHz and demonstrated its application in pneumonia image detection⁴. Furthermore, it is reported an all-optical computer with linear operations, nonlinear functions, and memory entirely in the optical domain, operating at over 100 GHz⁵. These advancements strongly suggest that 100 GHz optical computing is achievable in the future with our structure, which is the basis for our claim.

We sincerely appreciate the reviewer's observation that this section lacked sufficient supporting literature. We will incorporate this analysis and relevant references into the revised manuscript. Thank you again for your insightful comments.

Revision to Comment 4:

Thank you for pointing out the issues in our writing. After carefully reviewing the comments, we have made detailed revisions to the relevant sections of the manuscript. Specifically, we have updated the discussion section as follows:

The bandwidth of optical devices continues to improve. On-chip modulators have already achieved rates exceeding 110 GHz⁴⁶, while photodetectors are capable of

supporting signals up to 180 GHz⁴⁷. Owing to the utilization of high-bandwidth devices, optical computing benefits from inherently higher frequency, which has the potential to exceed 100GHz-double the current computation frequency.

Reviewer #1 (Remarks on code availability)

Comment:

The source file contains two parts of the code. The authors have shared both the code for designing the device and creating its layout, as well as the code for the generative AI model. The code of generative AI includes a README file and can be executed directly. However, the device design code lacks a README file. Upon opening the project, I found that running it requires a license from Luceda, please provide further details on the license.

Response to Comment:

Apologies for the earlier oversight. In the updated version of the source code, we have added a README file for the device design section to help guide users through the setup and usage process.

Please note that the device layout design is based on tools provided under a license from Luceda. This license grants access to Luceda's proprietary libraries, which are essential for running the project. These libraries include built-in functions for tasks such as waveguide drawing, layout generation, and photonic device parameterization. Users with a valid license can take full advantage of these resources to design and simulate custom photonic devices within the same framework.

Unfortunately, due to licensing restrictions, we are unable to redistribute Luceda's internal libraries directly. However, users affiliated with institutions or organizations that already hold a Luceda license should be able to run the project without issue. If you have trouble accessing the tools or need help obtaining a license, we would be happy to offer further guidance.

Revision to Comment:

We have added two README files to the codebase to guide researchers with a valid Luceda license in reproducing our results using the provided code:

“This project provides code for the design and optional simulation of an AWG-based photonic device using the Luceda IPKISS platform and the CSiP180A1 PDK. The design integrates an arrayed waveguide grating (AWG) with a microring resonator filter structure.

The workflow includes four configurable stages: generation, simulation, analysis, and finalization. Users can adjust design parameters such as channel spacing, waveguide width, and the free spectral range (FSR). Layouts are automatically visualized during the design process, and simulation results (if enabled) are saved for further analysis.

Environment Requirements:

Python 3.6+

Luceda IPKISS (with valid license)

NumPy

Custom modules: CSiP180A1, ipkiss3, rect_awg/, and importawg/

Simulation and analysis can be enabled via flags in the script. Output files are saved under a structured designs/ directory named according to the design parameters.

This code is intended for researchers and engineers working in silicon photonics design and AWG filter prototyping.”

“This project contains the code for designing and simulating an AWG-based photonic device using the Luceda IPKISS platform. It includes the generation of an arrayed waveguide grating (AWG), integration with microring resonators (MRRs), photodetectors (PDs), and Mach-Zehnder modulators (MZMs). The code supports design, optional simulation, result analysis, and finalization.

The script allows users to configure key parameters such as the number of channels, channel spacing, and waveguide width. Results including layout files and optional simulation outputs are saved under a structured folder based on the design parameters.

Environment Requirements:

Python 3.6 or higher

Luceda IPKISS (with valid license)

NumPy

Custom modules provided in the project (e.g., rect_awg, awg_generate, importawg)

Please note that a valid Luceda IPKISS license is required to run layout and simulation functionalities, as they rely on proprietary libraries. A README has been included to guide users through the device design flow.

This setup is intended for researchers and engineers working on silicon photonics and photonic computing system design.”

We are very grateful for your comments and valuable insights. Your feedback has provided important perspectives that helped us enhance both the technical depth and presentation of our work. We hope that our revisions have addressed your concerns.

Reviewer #2

General Comment:

The authors propose and experimentally demonstrate an optical cloud computing system within an edge-metro network. Their approach involves transmitting input data and model weights from the edge to optical computing nodes in the cloud, performing computations optically, and returning the results to the edge. Since data remains in the optical domain throughout the process, the system significantly reduces the power consumption of lasers. Their experiments demonstrate orders-of-magnitude lower energy consumption compared to conventional electronic cloud computing.

Strengths:

I appreciate the authors' effort to eliminate optical-electronic conversions and leverage optics for both communication and computation. This is a promising direction for low-power AI acceleration. The experimental methodology is well-documented, and the results provide strong initial validation of the concept. However, I have several concerns and questions regarding scalability, numerical precision, and system feasibility for modern AI workloads, as listed below.

Response to General Comment:

We sincerely appreciate the reviewer's positive feedback on our work and their recognition of the potential of optical cloud computing for low-power AI acceleration. Regarding scalability, we acknowledge the importance of evaluating the system's ability to support large-scale AI workloads. Our architecture is designed to leverage the inherent parallelism of optical computing, which can be extended to handle more complex models. We will provide additional discussion on this aspect in the revised manuscript.

For numerical precision, we understand that maintaining sufficient computational accuracy is crucial for practical deployment. We have conducted a detailed analysis of the precision required for different AI models and have demonstrated that our system has the potential to support modern generative AI models at a 10 GHz operation rate. Further improvements in precision are possible with optimized hardware, as we will elaborate in our response to specific comments. We appreciate the reviewer's insightful

questions and will address them in detail in our responses.

Comment 1:

Scalability & Multi-OPU Processing

From what I understand, the experimental results focus on a single convolution operation or a single convolutional layer. However, modern generative AI models rely on deep neural networks with multiple layers and millions to billions of parameters. The authors suggest that multiple OPUs could be used to process different layers, but this is not experimentally demonstrated. More details on the parallelization strategy would strengthen the paper. If multiple OPUs are used sequentially, does the optical signal require amplification after each layer? If so, would this introduce power and latency overheads that reduce the system's efficiency? Most deep learning architectures include non-linear functions (e.g., ReLU, GELU, Softmax) between layers. While there are examples of optical non-linearity in literature, they come with challenges such as signal amplification, integration of multiple materials, and packaging issues. How do the authors plan to integrate non-linear functions? Would this require switching back to electronic computation at certain stages? If a single layer has more parameters than one OPU can handle, how would intermediate results be stored or aggregated? Would this introduce additional latency or require hybrid optical-electronic processing?

Response to Comment 1:

We sincerely appreciate the reviewer's valuable feedback on the scalability of our proposed scheme. Below, we address the three key points raised by the reviewer, provide detailed responses to each concern, and highlight corresponding revisions made in the manuscript to reflect these improvements.

A. Details on the parallelization strategy

We fully agree with the reviewer's perspective that parallel strategies can enhance the performance of the proposed scheme. Parallelism has always been a critical component in large-scale model processing. In future cloud optical computing systems, it is expected to handle all parameters of large-scale models. Our proposed scheme inherently supports this potential, enabling extensive parallel computation and multi-layer processing while maintaining high scheduling flexibility. To provide a clearer illustration of the parallel computation mechanism in our proposed scheme, we have

added **Fig. R4** to the supplementary materials. This figure demonstrates two key aspects: (1) how multi-layer neural network operations are executed, and (2) how multiple optical processing units (OPUs) perform parallel computations to address the challenge of excessive computational parameters. In our design, the weights of a single convolutional kernel are loaded within two adjacent FSRs during one time slot. Leveraging wavelength-division multiplexing, multiple sets of weights can be loaded simultaneously within the same time slot. With different kernels assigned to separate wavelength ranges, multiple different convolution operations can be performed at the same time. In the cloud computing center, optical signals are routed to different OPUs based on their wavelengths to fulfill parallel computational requirements.

Fig. R4 Parallelization strategy in cloud optical computing. **a**, Weights are loaded across three dimensions (wavelength, frequency, and time), enabling multi-layer neural network computations and parallel processing. **b**, Multiple OPUs in the computing center execute distinct computational tasks. **c**, Weights and input data are transmitted from edge nodes to the computing center, while computational

outputs are continuously aggregated and returned to end-users. **d**, Outputs of parallel computations are sequentially delivered to edge nodes via frequency-division multiplexing, with different time slots corresponding to neural network outputs at distinct layers in **e**.

After completing one layer's computation, the convolutional kernels required for the next layer are transmitted via optical fiber in the subsequent time slot. The same OPUs then perform the next layer's computation using newly generated optical signals. It is worth noting that each layer's light source is independently generated at the edge node, ensuring both model security and OSNR consistency across all layers. This fully decoupled architecture eliminates physical dependencies between layers, thereby avoiding the need for additional optical amplifiers and power consumption. The uniform OSNR guarantees robust performance for deep neural network computations.

B. Optical Nonlinear Function

Nonlinear function nodes is a critical part in deep neural networks. While our research primarily focuses on linear computation in the model, the integration of nonlinear activation functions into our system remains a critical design consideration. Implementing on-chip nonlinearity has long been a significant challenge, and numerous approaches have been proposed to address this issue.

Nonlinear effects are inherently present in optoelectronic conversion processes. While minimizing such nonlinearity is a key goal in communication systems, optical computing can leverage these effects to implement nonlinear activation nodes. Previous studies have demonstrated the feasibility of utilizing nonlinearities in modulators^{2,3} (electro-optic conversion) or photodetectors^{4,5} (optoelectronic conversion) for optical computing. Among these approaches, the photodetector-based nonlinearity presents a more practical solution for our method. The nonlinear activation function can be integrated in the photodetectors without additional devices on chip. This approach offers a significant advantage by eliminating the need for additional optical amplifiers or electronic-domain operations, thereby maintaining system simplicity and energy efficiency.

C. Task Partitioning

Fig. R5 Schematic diagram illustrating the principle of mapping a two-dimensional matrix convolution to several one-dimensional convolutions.

When a single layer's computation cannot be handled by a single OPU, multiple OPUs are employed for parallel processing, as illustrated in Figure R3. When multiple convolutional kernels exist within a single layer, each OPU is assigned to process one kernel. These tasks are executed concurrently through internal synchronization, completing simultaneously before being propagated to the next layer. This approach eliminates the need for additional data storage or merging operations. For oversized convolutional kernels exceeding the processing capacity of a single OPU, dimensionality reduction is applied to partition the kernel into subtasks. As demonstrated in Fig. R5, this kernel segmentation strategy enables distributed computation across multiple OPUs while preserving algorithmic integrity. In this scenario, data aggregation is performed in the electrical domain. Since the output of the OPU is already in the electrical domain, no additional optoelectronic conversion is required. In scenarios requiring multi-OPU collaboration, real-time electrical-domain data aggregation is performed with the support of clock synchronization, eliminating the need for additional data storage. Critically, this approach introduces no extra latency or optical-electronic processing overhead at the computational level. However, when OPU resources are constrained and partitioned tasks need to be processed sequentially, additional latency becomes inevitable. Notably, the dominant latency arises from

computational task queuing rather than data read/write operations⁶. The overall latency is thus critically dependent on the task granularity and available OPU resources.

In conclusion, with sufficient OPUs, task partitioning and synchronization effectively eliminate additional latency and data storage requirements. Although electrical-domain data aggregation is employed, this strategy avoids supplementary optical-electronic processing.

Revision to Comment 1:

We have added the corresponding contents in the Results section in page 7:

“Within each frequency cycle, the system employs specific frequencies to transmit the signal, while others load the weights. Concurrent computation tasks are mapped to separate wavelengths, enabling parallelism through spectral resource partitioning. As illustrated in the figure, different convolutions within the same network layer are assigned to independent OPUs for parallel execution. The detailed parallelization methodology, including wavelength assignment protocols and OPU synchronization mechanisms, is comprehensively described in Supplementary Note 4.”

We have also added the corresponding contents in the Results section in page 8:

“When the scale of convolutional operations exceeds the processing capacity of a single OPU, the computation can be partitioned into smaller sub-convolutions and processed in parallel by multiple OPUs. For example, an image convolution task may be divided into multiple one-dimensional convolutions, each assigned to a dedicated OPU for simultaneous execution. A method to decompose image convolution into one-dimensional convolutions is proposed as shown in Supplementary Note 4, allowing the image convolution operation to be distributed across three OPUs for simultaneous computation.”

We have changed the name of Supplementary Note 4 and added the corresponding contents in this part in the supplementary material in page 7. We have also added the Fig. R4 as Fig. S6 in the revised version of the supplementary material.

“Supplementary Note 4: Parallelization strategy in cloud optical computing

In this framework, we propose the method to decompose complex computations into multiple simpler and relatively independent subtasks, which can be processed

simultaneously. This decomposition includes both the parallel execution of multiple convolutional operations in a large scale model and the dimensional reduction of large-scale convolutions into low-dimensional convolutions processable by individual OPUs. By harnessing the multidimensional parallelism across time, frequency, wavelength, and spatial domains, our architecture achieves hyper-dimensional parallel computation, enabling the handling of all parameters in large-scale models while maintaining high scheduling flexibility.

Within the design, the complete weights of a single convolutional kernel are loaded within one frequency cycle during a single time slot, as shown in Fig. S6. Leveraging wavelength-division multiplexing, multiple weight sets can be simultaneously loaded across distinct spectral channels. By assigning different kernels to separate wavelength ranges, different convolution operations are executed concurrently. In the cloud computing center, optical signals are dynamically routed to dedicated OPUs based on their wavelengths to fulfill parallel computational demands. A distributed array of OPUs is strategically deployed in the cloud optical computing center to support high-speed edge computations. As shown in the figure, results are generated in the computing center and transmitted back to the edge with ultralow latency. This architecture ensures minimal overhead through two key mechanisms: (1) electrical-domain data aggregation under clock synchronization eliminates the need for additional optoelectronic conversions, and (2) task partitioning and OPU coordination avoid redundant data storage. While constrained OPU resources may introduce queuing delays for sequential subtasks, latency remains dominated by computational queuing rather than data I/O operations. Regarding the implementation of nonlinear layers, the approach most optimally aligned with our architecture leverages the inherent nonlinearity of photodetectors, a strategy has been widely studied in prior works^{1,2}. This method enables the integration of nonlinear activation functions without introducing additional device complexity or power overhead. However, given the diversity and precision requirements of nonlinear activation functions (e.g., ReLU, sigmoid), implementing these functions in the electrical domain may serve as a viable alternative. While this introduces a trade-off between photonic integration density and computational

flexibility, it ensures compatibility with conventional deep learning frameworks and facilitates precise activation shaping through programmable electronic circuits.

Given that convolutional tasks may exceed the processing capacity of a single OPU, this work further investigates the partitioning of complex convolutions into low-dimensional sub-convolutions for distributed execution across multiple OPUs. This task decomposition strategy enables efficient processing of the image datasets presented in this study, achieving scalable performance while preserving computational accuracy.”

Comment 2:

Feasibility for Modern AI Models

The MNIST digit classification task is relatively simple and can be performed with very low computational precision. Additionally, the GAN architectures used (from 2016-17) are significantly smaller and less complex than today's generative AI models (diffusion models, transformers). How well do these architectures represent real-world AI workloads in edge devices? Could the proposed system handle state-of-the-art generative AI models? What modifications would be required to scale this system to modern deep-learning architectures?

Response to Comment 2:

Thank you for your insightful question. We acknowledge the reviewer's observation that the GAN architectures used in this paper are indeed smaller and less complex than state-of-the-art generative AI models. However, we believe that the experiments conducted in this work effectively capture the key functionalities representative of real-world AI workloads. Although our paper experimentally demonstrates relatively simpler models, these are sufficient to validate the capability of our computational system to perform diverse computational tasks. In the main text, we explicitly demonstrate fundamental arithmetic operations, including addition, subtraction, and multiplication. Furthermore, by utilizing wavelength-division multiplexing, we successfully implemented the loading of negative-valued weights. This capability covers the foundational linear computational operations required by neural networks. For state-of-the-art generative models, the primary hardware requirements for the optical computing focus on adequate computational precision, computational efficiency, and scalability. Therefore, hardware platforms that can efficiently perform large-scale computations with sufficient precision inherently fulfill the requirements of advanced generative AI models. Below, we will elaborate further on these three key aspects: computational efficiency, computational scalability, and computational precision.

Computational Scalability

Computational scalability is demonstrated through two primary aspects: the parallelization across multiple computational units and the expandability of individual

computational units. In advanced generative AI research, as researchers continually explore the limits of generative AI, the complexity of network architectures increases significantly, demanding greater computational depth and breadth. Effectively addressing this increased computational breadth—particularly how to handle larger-scale computations—has become an important research focus. A promising approach to meet these growing computational demands involves scaling up individual computing units while also implementing parallel computing strategies through multiple units. In our previous discussion, we have detailed how individual OPUs can operate in parallel within our proposed architecture. Specifically, leveraging WDM technology, multiple sets of weights can be simultaneously loaded within the same time frame. By allocating distinct convolutional kernels to separate wavelength bands, multiple convolution operations can be executed concurrently. In a cloud computing center, optical signals can be efficiently routed to different OPUs according to their wavelength assignments, finally facilitating parallel computation.

To expand the capability of an individual OPU itself, the primary strategy involves increasing the number of input and output channels in the AWGR structure.

Fig. R6 Design size of AWGR for different wavelength spacings and channel numbers. From bottom to top: 4, 8, 16, 32, and 64 channels.

Fig. R6 shows AWGR structures with varying scales, specifically 4, 8, 16, 32, and 64 channels. For a 64-channel AWGR, up to 64 channels of signal can be simultaneously

input, with each channel capable of accommodating 64 wavelengths within a single FSR. Consequently, the number of simultaneous computations scales quadratically with the number of channels, indicating that the computing capacity can increase significantly, as shown in Fig. R7. This clearly demonstrates the substantial potential of our architecture for scaling computational workloads.

Fig. R7 Computational speed as a function of device size with a, 50G AWGR; b, 100G AWGR.

Computational Efficiency

As shown in our paper, our hardware demonstrates superior computational efficiency. Compared with conventional fully electrical cloud computing platforms, our optical-computing architecture offers up to two orders of magnitude improvement in computational efficiency. This significant advantage arises from the inherent nature of optical computing, where data transmission inherently coincides with computation. In the architecture illustrated in Fig. R4, increasing the computational scale by adding more computational nodes—without altering the size or complexity of individual nodes—does not affect the overall computational efficiency. However, expanding the scale of a single computational node itself will influence computational efficiency, as the relationship between the computational rate of an individual node and its total computational scale is non-linear. This relationship is clearly demonstrated in **Fig. R8**. From the figure, it can be observed that increasing the OPU scale enhances computational efficiency. However, as the scale continues to grow, the computational efficiency of an individual OPU gradually approaches a limit of approximately 15 mW/TOPs. This represents an improvement of approximately one order of magnitude compared to the values previously reported in our paper of 118.6mW/TOPs. This

indicates that excessively enlarging the scale of a single OPU is unnecessary and may not yield proportional efficiency benefits. In practical applications, maintaining each OPU within a moderate size range—such as 32 or 64 channels—and employing parallel computing and task decomposition to leverage multiple OPUs simultaneously, is a more suitable and efficient strategy. Notably, operating individual nodes with a suitable wavelength number yields greater overall computational efficiency. Consequently, when faced with larger-scale computational demands, our architecture naturally achieves higher computational efficiency, demonstrating its suitability and adaptability for advanced generative AI workloads.

Fig. R8 The computational efficiency varies with changes in the AWGR size and the number of wavelengths employed. Generally, a larger AWGR scale can achieve higher computational efficiency, provided the number of wavelengths utilized remains within an optimal range. Beyond this range, the efficiency gains diminish. Additionally, the incremental improvement in efficiency resulting from continuously increasing the AWGR scale exhibits diminishing returns due to marginal effects.

Computation precision

In our experiments, these computational operations consistently demonstrated a computing precision of approximately 7 bits. Notably, this precision was primarily limited by the experimental equipment rather than by the intrinsic capabilities of our proposed architecture. Indeed, optical computing has already been demonstrated to achieve computational precisions exceeding 9 bits⁷, which is sufficient to support state-

of-the-art large-scale generative models. Furthermore, recent studies have widely shown that advanced generative models, including large language models, can operate effectively at 8-bit precision or even lower^{8,9}. We validated the performance variation of the Denoising Diffusion Implicit Model (DDIM) in image generation with computational precision using the model presented in Fig. R9.

Fig. R9 The structure of the denoising diffusion implicit model used in this paper.

Building upon the original DDPM model, DDIM introduces a leapfrog sampling strategy and a variable variance mechanism, which significantly reduces the sampling time of diffusion models, facilitating rapid evaluation of model generative performance at different computational precisions¹⁰. For this purpose, we utilized the official open-source implementation and its pre-trained model trained on the CelebA dataset. We introduced quantization noise in key layers of the DDIM model, including convolutional layers, attention mechanisms, and normalization blocks, to simulate the variation in model generation effects under different computational precision bit widths. The specific parameters were set as: timestep=100, eta=0. By using the Fréchet Inception Distance (FID) metric, we compared the performance of the model at different quantization precisions, as shown in Table R3.

Table R3 Performance of the DDIM at different precisions

Precision	4 bits	6 bits	8 bits	32 bits
FID	185.48	101.82	54.23	48.15

The results indicate that with the aid of advanced quantization algorithms, even a low

precision can support the generative performance of the model. This efficient computation at low precision provides a feasible implementation path, further validating the feasibility of optical computing chip systems in handling large-scale, complex generative AI models. Thus, through the synergistic optimization of hardware and quantization algorithms, it is expected that this system can efficiently operate larger scale, more complex modern deep learning models on edge devices.

Overall Analysis

Firstly, as shown earlier, from both computational architecture and computational scalability perspectives, our system inherently possesses the capability to process the extensive data flows characteristic of advanced generative AI models. However, the substantial data requirements of such state-of-the-art AI exceed the practical data-handling capacity currently achievable in the laboratory. This limitation primarily arises due to the large time-consuming of data import and export processes in experimental setups. To address this, we experimentally characterized key device parameters and subsequently employed these experimentally derived parameters in numerical experiment. These confirm our device's capacity to handle the computational demands of advanced generative AI models under realistic operational conditions. Our proposed system is demonstrated to be capable of handling the DDML, as shown in **Fig. R10**.

Fig. R10 The impact of computational precision on the output results in diffusion models.

The results demonstrated the performance of output images under 8-bit and 32-bit computational precision. The reduction in Fréchet Inception Distance (FID) from 54 to 48 highlights the enhancement in image generation quality with higher precision, as

evidenced by the improved visual quality of the generated face images. The numerical experiments conducted with this model yielded an FID value of 54.31, closely aligning with the performance at 8-bit precision. This outcome is partly attributed to our device being validated for 7-bit computational accuracy during testing, and partly due to the tolerance of advanced deep networks to lower precision, which allows our computational results to approximate those at 8-bit precision. These results collectively demonstrate the applicability of the proposed architecture to diffusion-based models. As discussed, the key distinction between modern large-scale models—whether diffusion or transformer architectures—and the GAN-based network evaluated in our experiments lies predominantly in their scale. However, their fundamental reliance on core operations such as convolution and matrix multiplication remains unchanged, and thus their hardware requirements are intrinsically aligned, which is supported by our architecture.

To effectively process advanced deep learning networks, the most critical future enhancement involves integrating nonlinear activation functions directly onto the chip. The ability to perform all computational operations optically can significantly minimize the frequency of optical-electrical-optical conversions and significantly enhance computational efficiency. Nonlinear activation functions represent essential components within deep neural networks. Although our current research primarily emphasizes linear computational operations, integrating nonlinear activation capabilities remains a fundamental consideration in our system's future design. On-chip implementation of nonlinear functionalities has long posed a considerable challenge, prompting numerous studies and proposed solutions, the nonlinear activation achieved via photodetectors exhibits a particularly practical approach for our method. Such integration allows nonlinear functions to be implemented within photodetectors without requiring additional on-chip devices. This approach significantly simplifies system architecture by eliminating supplementary optical amplifiers or electronic-domain operations, thereby preserving system simplicity and maximizing energy efficiency.

Revision to Comment 2:

To demonstrate the scalability of computational capacity for an individual OPU, we provide a more detailed analysis in the supplementary material, focusing specifically on the scalability of the OPU and the corresponding impact on chip size during this expansion process. We have added the contents of Fig. R1 and Table R1 in Supplementary Note 1 as Fig. S2 and table S1, in the revised version of the supplementary information. We have also added the corresponding texts regarding the description of these figures in Supplementary Note 1:

“Scalability of the AWGR-based processing unit

The approach of using an AWGR for convolution computations has a limitation in that the size of the device is large and, theoretically, difficult to significantly reduce. By combining Equations (10) and (12), it can be determined that:

$$R = \frac{n_s d_i d_a \Delta \lambda_{FSR}}{2 \lambda \lambda_0} \quad (14)$$

This indicates that increasing the number of channels will significantly enlarge the device size due to the expanded FSR. The specific impacts are illustrated in Fig. S2. Fig. S2a shows the AWGRs with a 50 GHz channel spacing at varying channel numbers, while the Fig. S2b displays the corresponding dimensions for a 100 GHz channel spacing. From the figure, it can be observed that as the channel number increases, the device size grows accordingly. This scaling is primarily driven by the elongation of the short side caused by the expanded star coupler structure. Notably, even in a 64×64 configuration, the short side length remains below 1.8 cm. Table S1 quantifies the impact of channel number scaling on device size and peak computational capacity under a fixed channel spacing of 100 GHz. As the number of channels increases from 4 to 64, the device size scales by 400%, while the computational speed exhibits a remarkable 25,500% enhancement. This results in a significant improvement in computational efficiency per unit area. Although enlarging the AWGR introduces a moderate increase in physical footprint, the substantial gains in computational throughput justify this trade-off, demonstrating the scalability and practical viability of the proposed architecture.”

For state-of-the-art models, the rapidly increasing power consumption associated with expanding computational scale and user demands has become a key factor limiting further growth. To address this issue, we have added a detailed analysis in the *Discussion* section of the main manuscript, examining the anticipated power consumption trends of our proposed architecture in future scenarios. This analysis demonstrates the notable computational efficiency of our architecture when handling large-scale computational workloads. We have added the corresponding texts in *Discussion* section of the main manuscript in page 14:

“It is noteworthy that as the computational scale of OPUs increases, power consumption scales linearly with component size, while the maximum computational rate scales quadratically^{43,44}. Therefore, computational efficiency can be further enhanced with increased component scale, which is further discussed in the Supplementary Note 3.”

To provide readers with a clearer understanding of this point, we have included an additional detailed analysis in Supplementary Note 3. Correspondingly, Fig. R8 has also been added to the Supplementary material as Fig. S5:

“Beyond the superior efficiency analyzed in the main manuscript, this architecture offers the additional advantage that, when scaling up to meet the demands of larger computational tasks, the power efficiency can be further improved, as demonstrated in Fig. S5. The computational efficiency of an individual OPU gradually approaches a limit of approximately 15 mW/TOPs, which represents roughly a one-order-of-magnitude improvement compared to the previously reported value of 118.6 mW/TOPs. By selecting an optimal number of wavelengths for each node, the overall computational efficiency can be the best. Consequently, our architecture naturally achieves superior computational efficiency under increased computational demands, highlighting its substantial suitability and adaptability for advanced generative AI workloads.”

To further explore the impact of computational precision in optical cloud computing on more advanced models, we conducted an in-depth analysis using the model shown in Fig. R9, specifically examining the output performance of the DDIM at various precisions. This investigation confirmed the capability of optical cloud computing to

support sophisticated generative model architectures. Fig. S28 has been added to the supplementary materials, and relevant discussions are included in Supplementary Note 8:

Fig. S28 The impact of computational precision on the output results in DDIM with the aid of QNCD

“To further explore the potential of optical computing for advanced and complex models, additional research has been conducted to determine whether the precision of optical computing can adequately support modern deep neural networks. Direct application of the post-training quantization method struggles to meet the requirements of large models under relatively low-precision operations. In response, many advanced algorithms^{8,9,11,12} proposed to achieve low-loss quantization of large models. These quantization algorithms have typically been validated at 4, 6, and 8 bits, with the lowest quantization precision reaching 1-2 bits. Therefore, considering advanced quantization algorithms can significantly enhance the efficiency and performance of large model quantization computations. We employed the QNCD quantization method¹³ to test the DDIM model. The results, as shown in Fig. S28 indicate that with the aid of QNCD, even at lower bit widths (such as 8 bits), the generative performance of the model is largely preserved, which is similar to the output performance supported by our device.

This efficient computation at low bit widths provides a feasible implementation path, further validating the feasibility of optical computing chip systems in handling large-scale, complex generative AI models. Thus, through the synergistic optimization of hardware and quantization algorithms, it is expected that this system can efficiently operate larger scale, more complex modern deep learning models on edge devices. These advancements significantly bolster our confidence in optical cloud computing, leading us to believe that it will integrate more closely with people's lives in the future and provide a foundation for the widespread application of generative models.”

Nonlinear activation functions, an indispensable component of neural networks, are crucial for future applications if implemented optically. We have included discussions on this topic in Supplementary Note 4:

“Regarding the implementation of nonlinear layers, the approach most optimally aligned with our architecture leverages the inherent nonlinearity of photodetectors, a strategy has been widely studied in prior works^{1,2}. This method enables the integration of nonlinear activation functions without introducing additional device complexity or power overhead. However, given the diversity and precision requirements of nonlinear activation functions (e.g., ReLU, sigmoid), implementing these functions in the electrical domain may serve as a viable alternative. While this introduces a trade-off between photonic integration density and computational flexibility, it ensures compatibility with conventional deep learning frameworks and facilitates precise activation shaping through programmable electronic circuits.”

Comment 3:*Numerical Precision & Data Representation*

The reported 7-bit integer precision may be sufficient for MNIST classification, but modern generative AI models typically use low-precision floating-point formats (e.g., FP8, FP16) to preserve dynamic range. Integer representation is known to struggle with dynamic range, especially in generative tasks. If higher precision is needed, what modifications would be required to support it?

Response to Comment 3:

We sincerely appreciate the reviewer's insightful question. The computational precision indeed plays a crucial role in determining the practical applicability of the proposed architecture. As shown in Fig. R11, increasing computational precision enhances neural network performance. Our results indicate that for the model presented in our work, 7-bit precision is acceptable. For modern generative AI, high-precision cloud computing is needed to support models that are highly sensitive to noise. On the other hand, many studies have demonstrated that certain large language models exhibit high tolerance to lower precision^{8,9}. Our proposed architecture can support these models at a 10 GHz operation rate with the 7-bit precision, while some models can even function effectively with 6-bit precision. Therefore, while precision is important for accommodating a broader range of models, our current results are enough to demonstrate that the proposed architecture is sufficiently precise for implementing modern generative AI models.

In our experiments, we have also attempted to further enhance the achievable computational precision but were ultimately limited by the experimental equipment. The high-speed DAC available in our lab (operating at a 120 GSa/s sampling rate) supports a maximum of 8-bit precision. Considering the inevitable noise introduced in the experimental setup, 7-bit precision emerged as the highest practically achievable precision in our experiments.

However, 7-bit is by no means the maximum precision achievable with our proposed architecture. Notably, high-precision DACs operating at lower speeds can support 12-bit signal generation at a 12 GSa/s sampling rate¹⁴. This represents the most

fundamental approach to improving precision in our current experimental setup. Once the precision of electronic components is achieved, minimizing system noise becomes crucial. At this stage, optimizing the bandwidth of the components plays a vital role. A flat in-band response can help improve the system's SNR, thereby reducing the impact of noise on the received signal.

At the chip-design level, further advancements remain necessary. The on-chip electrical circuitry is critical to overall bandwidth performance, involving two primary factors. For the single-channel, circuit layout directly influences the transmission bandwidth and, consequently, the SNR under high-speed computations¹³. In multi-channel configurations, electromagnetic crosstalk between channels can further degrade SNR, ultimately limiting signal fidelity.

In addition, the electrical packaging of the chip requires continued optimization. Off-chip circuit design, including interconnect geometry and bonding schemes, can substantially impact signal precision. Recent efforts have explored three-dimensional integration approaches¹⁴, where photonic–electronic co-packaging has led to notable improvements in SNR, thereby enhancing computational accuracy at the system level.

Fig. R11 The performance of the network's output under varying computational precision levels.

Revision to Comment 3:

We have analyzed potential approaches to further improve the computational precision of the system. This has been added to the *Discussion* section on page 13 of the main manuscript:

“The lack of an available AWG that offer both high speed and precision represents a significant limitation in the experiment. This ultimately limits our achievable precision to within 8-bit. If a higher-performance AWG, such as one supporting 12-bit precision, is used, the computational accuracy can be further improved. Additionally, the overall bandwidth limitation of the system, primarily attributed to the modulator and PDs, plays a crucial role. The relationship between precision and baud rate is depicted in Fig. 4c, while the frequency response of the entire computing system is elaborated upon in Supplementary Note 8. Specifically, four Mach-Zehnder Modulator (MZM, T.MXH1.5) and a 100G photodetector (XPDV4121R-WF-FP) were utilized in this experiment, with the bandwidth of the MZM identified as the predominant factor affecting precision. The packaging process also affects precision by influencing the overall system bandwidth. This is closely related to the spacing and arrangement of high-frequency interconnections. The performance of the packaged device is presented in Supplementary Note 8.”

Thank you again for your detailed review and thoughtful comments. We have responded to each of your points with great care, and we hope that the revised manuscript meets your expectations. We truly appreciate your help in strengthening the clarity and rigor of our research.

Reviewer #3

General Comment:

This work proposed and demonstrated a seamless optical cloud computing system across edge-metro network by deploying optical computing nodes in the cloud, enabling direct access to OPUs through the existing optical network infrastructure from the edge. The manuscript is well-written, and the idea is interesting and innovative. Below, I have detailed my concerns and suggestions, which I hope will be useful for the authors to improve this manuscript.

Response to General Comment:

We sincerely appreciate the reviewer's positive feedback on our work and the recognition of the innovation and feasibility in the optical cloud computing. We also greatly appreciate the reviewer's insightful concerns and suggestions. We will carefully address each point and incorporate necessary revisions to further enhance the clarity, completeness, and impact of the manuscript. Thank you for your constructive feedback, which is invaluable in improving our work.

Comment 1:

As the authors mentioned in the paper, only one convolutional layer was conducted in the optical domain during the experiment, while all other operations were performed on a computer. My concern is that this cannot unlock the full potential of optical cloud computing. Particularly, the optical convolution process by the current method is relatively straightforward. I would expect multilayer convolution in the optical domain. Without this, I cannot recommend this paper to be published in Nature Communications.

Response to Comment 1:

We appreciate and agree with the reviewer's observation. If only a single layer of computation is implemented in the optical domain in real-world applications, it would indeed fail to fully leverage the advantages of optical computing—particularly its speed and efficiency. To address the reviewers' concerns, we conducted multi-layer neural network computations in the optical domain. The goal was to verify both the feasibility of implementing multi-layer operations within the proposed architecture and the validity of our experimental setup.

We propose an optical cloud computing framework based on independent computational nodes capable of performing various convolutional operations. In our framework, the structure of each neural network layer is decomposed into individual convolutional computation tasks, as shown in Fig. R12. Therefore, the position of the convolution task within the neural network makes no difference for our experiment. From the perspective of the OPU, it simply needs to process the received convolutional task and output the results.

Fig. R12 Parallelization strategy in cloud optical computing. **a**, Weights are loaded across three dimensions (wavelength, frequency, and time), enabling multi-layer neural network computations and parallel processing. **b**, Multiple OPUs in the computing center execute distinct computational tasks. **c**, Weights and input data are transmitted from edge nodes to the computing center, while computational outputs are continuously aggregated and returned to end-users. **d**, Outputs of parallel computations are sequentially delivered to edge nodes via frequency-division multiplexing, with different time slots corresponding to neural network outputs at distinct layers in **e**.

If the reviewer's concern were centered on how our proposed scheme could support a fully optical implementation of multi-layer networks, we apologize for not providing a more detailed explanation of our multi-layer architecture. Our approach is based on an electrical-optical-electrical structure to realize multi-layer processing, with the goal of decomposing the whole computing task into multiple simple computation processes rather than integrating all layers on a single chip. As illustrated in Fig. R12, the idea of our scheme is to decompose a complex model into multiple simple tasks, each of which can be independently executed by a single OPU.

Through our experiments, we have demonstrated the following key points: 1. The

proposed OPU can function equivalently to a remote computing node with 7-bit precision. 2. Our proposed computing architecture is able to support various computing tasks. 3. The OPU is capable of being applied to practical tasks involving generative models. We will include a clearer description of this architecture and its operation flow in the revised manuscript to avoid potential misunderstandings and better communicate the flexibility and scalability of our proposed system. In our method, after completing one layer's computation, the convolutional kernels required for the next layer are transmitted via optical fiber in the subsequent time slot. The same OPUs then perform the next layer's computation using newly generated optical signals. It is worth noting that each layer's light source is independently generated at the edge node, ensuring both model security and OSNR consistency across all layers. This fully decoupled architecture eliminates physical dependencies between layers, thereby avoiding the need for additional optical amplifiers and power consumption. The uniform OSNR guarantees robust performance for deep neural network computations.

We also considered the reviewers' concern about the potential decrease in output precision after computations through multiple layers of an optical computing network. This is indeed a valid concern, as optical computing, being an analog computation method, generally experiences a significant drop in precision compared to electronic computing. To address this confusion and respond to the reviewer's query, we processed the multi-layers of a seasonal transfer task in optical domains. Some of the error distribution for the results obtained from the hidden layer is shown in **Fig. R13**. We display images numbered from 0 to 15, which exhibit varied and non-uniform error distributions.

Fig. R13 Seasonal transformation task output from the hidden layer after optical domain computation: Output nodes 0 to 15

The results confirm that the quality of the output images did not significantly change after multi-layers of optical domain computation. After two layers optical-domain computation, the LPIPS metric slightly degraded from 0.0201 to 0.0210, while the FID increased from 11.64 to 13.09. Given the inherent variability of these metrics, a single image performance cannot accurately demonstrate output performance. Overall, the multi-layer optical domain computation exhibits performance similar to the single-layer optical computation demonstrated in the paper, generally equivalent to 7-bit quantization accuracy with LPIPS and FID of 0.0224 and 16.33. This aligns with our experimental method described in the previous manuscript, where the first layer is computed in the optical domain, and subsequent layers are simulated in the electronic domain with 7-bit quantization to mimic the optical computation mode.

The output results are displayed in Fig. R14 to provide reviewers with a more intuitive understanding of the impact of the proportion of optical computation in the overall experiment on the output results. The visual perception of the images and the displayed metrics are similar. The simulation results at 6-bit precision are clearly inferior to those of optical computing. The final output results by computed in optical domain are similar to those from the all electrical network output at 7-bit quantization and slightly inferior to the 8-bit quantization results. The best output performance is observed at 32-bit quantization.

Fig. R14 Output results of seasonal transformation under different computing environments

To address the reviewer’s concern regarding the absence of multi-layer optical-domain computations in our initial submission, we have provided the following clarifications and results updates. First, we have elaborated on the computational capabilities of our proposed photonic processing unit and the overall architecture of the optical cloud computing system. We emphasize that, from the perspective of a unitary computational device, the layer from which a given task originates does not affect the execution of the convolution operation itself.

Second, we have introduced a new schematic illustrating how complex generative tasks are decomposed into parallel computing workloads across multiple OPUs within our architecture. This diagram serves to clarify the envisioned operational flow of our system when handling deep, multi-layered neural networks. This also demonstrates the

consistency in the underlying computational logic between multi-layer and single-layer operations.

Finally, to directly address the reviewer's inquiry on multi-layer optical computation, we conducted a seasonal image transformation task in which multi-layers of the model were computed in the optical domain. The experimental results show that the image quality generated after multi-layers of optical-domain computation is comparable to that of single-layer optical computation and aligns closely with the output from a 7-bit quantized digital simulation. These results are supported by both visual comparisons and quantitative metrics.

In summary, we have addressed the reviewers' requests and incorporated detailed responses and additional results in both the main manuscript and the supplementary materials. We sincerely thank the reviewers for this constructive suggestion, which have helped improve the quality of our work.

Revision to Comment 1:

We have changed the name of Supplementary Note 4 and added the corresponding contents in this part in the supplementary material in page 7. We have also added the Fig. R12 as Fig. S6 in the revised version of the supplementary material.

“Supplementary Note 4: Parallelization strategy in cloud optical computing

In this framework, we propose the method to decompose complex computations into multiple simpler and relatively independent subtasks, which can be processed simultaneously. This decomposition includes both the parallel execution of multiple convolutional operations in a large scale model and the dimensional reduction of large-scale convolutions into low-dimensional convolutions processable by individual OPUs. By harnessing the multidimensional parallelism across time, frequency, wavelength, and spatial domains, our architecture achieves hyper-dimensional parallel computation, enabling the handling of all parameters in large-scale models while maintaining high scheduling flexibility.

Within the design, the complete weights of a single convolutional kernel are loaded within one frequency cycle during a single time slot, as shown in Fig. S6. Leveraging wavelength-division multiplexing, multiple weight sets can be simultaneously loaded

across distinct spectral channels. By assigning different kernels to separate wavelength ranges, different convolution operations are executed concurrently. In the cloud computing center, optical signals are dynamically routed to dedicated OPUs based on their wavelengths to fulfill parallel computational demands. A distributed array of OPUs is strategically deployed in the cloud optical computing center to support high-speed edge computations. As shown in the figure, results are generated in the computing center and transmitted back to the edge with ultralow latency. This architecture ensures minimal overhead through two key mechanisms: (1) electrical-domain data aggregation under clock synchronization eliminates the need for additional optoelectronic conversions, and (2) task partitioning and OPU coordination avoid redundant data storage. While constrained OPU resources may introduce queuing delays for sequential subtasks, latency remains dominated by computational queuing rather than data I/O operations. Regarding the implementation of nonlinear layers, the approach most optimally aligned with our architecture leverages the inherent nonlinearity of photodetectors, a strategy has been widely studied in prior works^{1,2}. This method enables the integration of nonlinear activation functions without introducing additional device complexity or power overhead. However, given the diversity and precision requirements of nonlinear activation functions (e.g., ReLU, sigmoid), implementing these functions in the electrical domain may serve as a viable alternative. While this introduces a trade-off between photonic integration density and computational flexibility, it ensures compatibility with conventional deep learning frameworks and facilitates precise activation shaping through programmable electronic circuits. Given that convolutional tasks may exceed the processing capacity of a single OPU, this work further investigates the partitioning of complex convolutions into low-dimensional sub-convolutions for distributed execution across multiple OPUs. This task decomposition strategy enables efficient processing of the image datasets presented in this study, achieving scalable performance while preserving computational accuracy.”

Fig. R14 has been added into the supplementary materials in Supplementary Note 8 as Fig. S27. A detailed analysis of these results is provided on page 15 of the

supplementary document:

“In the aforementioned task, all convolution layers for handwritten digit recognition are implemented optically in the experiment. In contrast, for generative AI, only the first convolutional layer is processed in the optical domain, which is owing to the significantly larger model scale compared to the digit recognition task. To obtain enough results in our experiment, we began by numerically modeling our proposed architecture and employed this model to replace selected portions of the neural network during evaluation. This method integrates seamlessly with our proposed system. This is because our cloud optical computing architecture fundamentally aims to decompose a complex neural network computation into a sequence of individual operations. Within this framework, our device is employed to carry out discrete convolutional tasks. As a result, whether the computation originates from a single-layer or multi-layer network becomes essentially irrelevant in terms of how the operations are executed.

To verify whether the performance remains consistent after multi-layer optical domain computation, more analysis is conducted as illustrated in Fig. S27. The output results of the generative AI model were evaluated under settings with multi-layer optical-domain computations, in addition to different quantization noise levels in the electrical domain. A comparison of the results demonstrates a high degree of consistency in the outputs produced by optical-domain computations and the electrical domain with 7-bits. The results obtained from executing the first and second layers of the model in the optical domain demonstrate a performance level comparable to that of the electronic counterpart with 7-bit quantization. In the main manuscript, we examined the precision of fundamental operations and found that each maintained a computational accuracy of approximately 7 bits. This aligns closely with the results observed in Fig. S27. When only the first layer is computed in the optical domain, the model achieves the FID and LPIPS of 11.64 and 0.0201, separately. These metrics remain highly consistent after extending the optical computation to multi-layers. Specifically, the FID slightly degrades to 13.09, and the LPIPS increases marginally to 0.0210—both changes being minimal. Moreover, the simulated results obtained from optical-domain computation closely match those of the 7-bit quantized all-electrical baseline shown in the table. This

consistency holds for both the single-layer and multi-layer optical computing scenarios, indicating that the outputs from optical computation are comparable to those achieved under 7-bit quantization in electrical domain. These findings support two key conclusions: first, the inherent computational precision of optical-domain operations is approximately 7 bits; second, the performance of a network with a single optical layer can be extrapolated to networks with multiple optical layers.”

The following content has been added to the Results section on page 11 of the main manuscript:

“More detailed results are provided in Supplementary Note 8 to further validate the rationale of this approach.”

Comment 2:

In this paper, the computational accuracy, speed and energy efficiency of optical cloud computing are described in detail, and it is pointed out that optical cloud computing can achieve a variety of image generation tasks with low energy consumption, but the specific effects of task realization are not compared. Without the specific effects of the image generation task, and the lack of comparative data (such as the accuracy of the generated image, loss of structure, detail retention, etc.), it is difficult to fully evaluate the advantages and disadvantages of optical computing in real-world tasks. It is suggested that comparative experiments of one or more generation tasks can be added in this paper to compare the performance of optical cloud computing and electronic cloud computing in the quality, speed, accuracy and stability of image generation.

Response to Comment 2:

We sincerely appreciate the reviewer's valuable suggestion regarding the comparative evaluation of image generation tasks. Our primary focus in this work is to demonstrate the computational accuracy, speed, and energy efficiency of optical cloud computing, highlighting its potential for AI acceleration. While we have shown that our system supports various image generation tasks with significantly lower energy consumption, we acknowledge that a direct comparison of generated image quality, structural fidelity, and detail retention with electronic cloud computing would provide further insights into its real-world performance.

To fully conduct a comparative evaluation of optical and electronic cloud computing technologies, we selected multiple tasks to assess the performance of our network in executing these tasks. We then compared the results obtained using our optical computing framework with the electrical cloud computing. For optical cloud computing, we employed our existing computational architecture, while for electronic cloud computing, we simulated the computations using the GeForce RTX 3090Ti. It is important to note that the communication infrastructure in electronic cloud computing follows a commercially available architecture, which does not influence the computational results but does impact the overall power consumption. When considering only the computational power consumption, the actual energy usage of the

electronic computing scheme is expected to be higher than the values recorded in the manuscript.

We firstly conducted comparative experiments on tasks including object generation, aerial photo mapping, semantic segmentation, and image depth detection. The results are summarized in **Fig. R15**. To quantify performance, we selected Peak Signal-to-Noise Ratio (PSNR) and Learned Perceptual Image Patch Similarity (LPIPS) as key metrics, which effectively assess image quality, structural fidelity, and perceptual similarity across different computational methods.

PSNR is an objective metric for assessing image quality. Its core principle is to evaluate the fidelity of a generated image by computing the mean squared error between the original and the distorted image. The PSNR is calculated using the formula:

$$PSNR = 10 \cdot \log_{10} \left(\frac{Max^2}{Mean(x - y)^2} \right)$$

where *Max* represents the maximum possible pixel value, *x* denotes the label image, and *y* denotes the generated image. A higher PSNR value indicates that the generated image is closer in quality to the label. And the LPIPS is a deep learning-based approach for image quality assessment. It leverages a neural network that is trained to approximate human perceptual judgments, thereby providing a score that reflects the perceptual similarity between a generated image and its original counterpart. A lower LPIPS score indicates a higher degree of similarity. In our experiments, we employ the pre-trained AlexNet model¹⁵ to calculate the LPIPS score. The labels correspond to images generated by the GPU, which serves as a reference for evaluating the performance of electronic cloud computing. A PSNR greater than 30 indicates that the images generated in the optical domain exhibit a high degree of similarity to those computed in the electronic domain. For structural similarity assessment, LPIPS values typically range between 0 and 1 for structurally similar images. An LPIPS value of 0.2 or lower suggests a highly similar structure. Given the inherent randomness in neural network processing, variations in noise distribution can introduce slight randomness in the generated images. An LPIPS value around 0.2 is considered as a good outcome.

Fig. R15 Comparison of psnr and lpips metrics across different tasks in image generation

The comparison between optical cloud computing and electronic computing is presented in **Fig. R15**, where the results illustrate their semantic similarity to the full-precision outputs. To determine the precision level of optical cloud computing for these tasks, we include the results obtained using 7-bit precision electronic computing. The figure demonstrates that our optical computing system achieves comparable performance. The semantic similarity of the generated optical computing outputs is quantified by an average LPIPS value of 0.23. It confirms that the essential features and structures are well-preserved, as illustrated in **Fig. R15**.

Fig. R16 the inset (f) of Fig. 5 in the manuscript

We further compared the experimental results of the season transfer tasks presented in Fig. 5(f) with their corresponding noise-added and noise-free outputs. For the seasonal transfer experiment, we introduced Structural Similarity Index Measure

(SSIM) and Fréchet Inception Distance (FID) as evaluation metrics to quantitatively assess the impact of noise on image quality and structural fidelity. SSIM is used to measure the structural similarity between two images and is widely employed in image quality assessment. The SSIM index is given by:

$$SSIM(x, y) = \frac{(2\mu_x\mu_y + C_1)(2\sigma_{xy} + C_2)}{(\mu_x^2 + \mu_y^2 + C_1)(\sigma_x^2 + \sigma_y^2 + C_2)}$$

In this expression, x and y represent the original and generated images, respectively. The terms μ_x and μ_y denote the mean intensities of the images, σ_x^2 and σ_y^2 are the variances and σ_{xy} is the covariance between x and y . The constants C_1 and C_2 are small values introduced to stabilize the division. SSIM is a metric ranging from 0 to 1, where higher values indicate better performance. A value greater than 0.9 is generally considered good, indicating that the generated image closely resembles the reference image in terms of structure and perceptual quality. FID is another widely used metric for evaluating image quality. A lower FID score corresponds to better image quality, with an acceptable threshold typically considered to be 50. When FID is below 50, the generated images are regarded as having good quality, closely resembling label images in distribution and perceptual coherence. FID is designed to quantify the similarity between two sets of images by comparing the distributions of feature representations extracted from a deep neural network. Specifically, it measures the Fréchet distance between the feature distributions of real and generated images. The FID is computed as follows::

$$FID(x, y) = \|\mu_x - \mu_y\|^2 + \text{Tr}(\sigma_x + \sigma_y - 2\sqrt{\sigma_x\sigma_y})$$

Here, μ_x and σ_x are same as above. Tr denotes the trace of a matrix. Typically, these features are extracted from an intermediate layer of the Inception network. In our study, we utilize the pre-trained Inception_v3 model ¹⁶ to compute the FID score.

Table R4 Performance comparison of metrics in the season transfer task

Index	Type	PSNR(dB)	SSIM	FID
Season Transfer	Electrical	31.77	0.9427	24.66
	Optical	30.63	0.9226	39.8

The performance of the season transfer tasks are shown in **Table R4**. A total of 12 images were processed for style transformation, with six images converted from winter to summer and six images converted from summer to winter. The SSIM and FID validate that optical cloud computing achieves performance comparable to that of advanced electronic computing networks for this task.

Table R5 Comparative Analysis of Performance Metrics in Optical vs. Electronic Cloud

Computing						
Index	Power Efficiency	Speed/core	FID	LPIPS	SSIM	Std
Optical cloud computing	118.6 mW/TOPs	3.6Tops	39.8	0.23	0.9226	0.0218
Electronic cloud computing	20.6 W/TOPs	0.06Tops	24.66	0.13	0.9427	0.0229

We present a comprehensive comparison in **Table R5**, where the power consumption and computation speed are based on the performance of the H200 GPU, which, according to our research, is the most advanced GPU available at the time of our experiments. The H200 GPU consists of 528 tensor cores, supporting a total computational throughput of 34 TOPs, with each core providing an average of 0.06 TOPs. In comparison, optical computing offers higher per-core speed of 3.6 TOPs and greater computational efficiency of 118.6 mW/TOPs. However, the current optical cloud computing technology is still constrained by precision limitations, preventing it from reaching the performance level of state-of-the-art electrical cloud computing. Improving the precision of optical cloud computing remains a key challenge and will be a decisive factor in determining whether it can truly replace electrical cloud computing in the future. FID, LPIPS, and SSIM respectively capture different aspects of image evaluation in diverse tasks: FID reflects the overall quality of generated images, LPIPS measures the preservation quality of semantic features, and SSIM assesses structural similarity. These metrics are employed to evaluate both the quality and accuracy of optical computing. In addition, we calculated the standard deviation of

SSIM values across 12 season-transfer output images to assess the stability of optical computation. Overall, the results show that the primary advantages of optical computing lie in its power efficiency and processing speed, while the quality of image generation tends to lag slightly behind that of electronic computation. In terms of computational stability, optical and electronic approaches perform comparably.

In our newly added work, we conducted extensive computations to obtain additional data that directly address the concerns raised by the reviewer regarding the quality, speed, accuracy, and stability of image generation. These issues have been addressed point by point in the responses above.

Revision to Comment 2:

We have conducted additional experiments in the manuscript to compare the performance differences between optical cloud computing and electronic computing. A comprehensive comparison was carried out from four perspectives: quality, speed, accuracy, and stability. The specific revisions and additions are detailed as follows.

We have revised the Fig. 5 in *Result* section in the main manuscript in page 11:

Fig. 1 experimental result of classification and generative AI tasks. a, The accuracy of MNIST handwritten digital image classification with different precision. b, Confusion matrix for MNIST handwritten digital image classification. c, Architecture of optical cloud computing system adapted for various tasks. d, Convolved waveforms from the first layer of the map edge detection task, with the red and blue lines representing the ideal and experimentally generated waveforms, respectively. e, Performance comparison between simulation and experiment with 6 different generative AI tasks (edges2handbags, edges2portrait, edges2shoes, map2edges, pix2pix-depth and segmentation). f, Image generation results for different tasks, season transfer (winter to summer; summer to winter) and semantic segmentation.

We have also performed additional analysis on the experimental results of the optical cloud computing of MNIST handwritten digital image classification and performance comparison between simulation and experiment in the revised manuscript in page 11-12:

“The impact of the limited precision of the OPU on the accuracy of MNIST handwritten digital image classification is investigated in Fig. 1a. To verify the experimental performance of processing all convolutional layers optically, the network was

simplified to include only one convolutional layer, as shown in the Supplementary Notes Fig. S16. When the bit precision is less than 5, the accuracy of the network dramatically improves as precision increases, rising from 11% to 90%. Once the bit precision exceeds 5 bits, the accuracy stabilizes around 92%. These trends are confirmed by experimental results in which 100 images from the MNIST dataset are processed through the optical cloud computing system. In the experiment, the convolution layer was executed optically, while the remaining operations were conducted on a computer, maintaining the same bit precision as the photonic devices. An accuracy of 88% was achieved, as confusion matrix shown in Fig. 1b. This demonstrates that a single OPU can support handwritten digit recognition of An accuracy of 88% with computation speeds^{11,12} reaching $6 \times 6 \times 2 \times 50 \text{ GHz} = 3.6 \text{ TOPS}$.”
“The performance comparison between the experiment and simulation results is presented in Fig. 1e, demonstrating that the optical computing system achieves performance comparable to 7-bit precision electronic computing. Also, a selection of the generative AI results is displayed in Fig. 1f, with additional results available in the Supplementary Note 8.”

We also analyzed the performance of optical cloud computing in terms of accuracy, image quality, and stability, the following content is added in the *Discussion* section of the revised main manuscript in page 13:

“These results in optical cloud computing exhibiting lower performance compared to electronic computing, particularly in terms of accuracy, generated image quality, and stability. For relatively simple tasks such as handwritten digit recognition, the accuracy of optical cloud computing reaches 88%, which is lower than the 92% achieved by electronic cloud computing. For image generation tasks, compared to full-precision computing, optical cloud computing achieves SSIM, FID, and LPIPS values of 0.92, 39.8, and 0.23, respectively. These results indicate that while optical cloud computing demonstrates a reasonable level of similarity to state-of-the-art electronic computing, its performance is approximately equivalent to that of electronic computing at 7-bit precision. Additionally, the variance of the SSIM parameter for the seasonal transfer task was calculated to be 5.1×10^{-4} , which is higher than the 3.4×10^{-4} observed

in 7-bit precision electronic computing. This difference can be attributed to factors such as channel fluctuations in the optical system.”

Comment 3:

The paper mentions that the low energy consumption of optical cloud computing centers can significantly reduce the operating costs of cloud computing, thus promoting the development of generative AI. However, the hardware of optical computing systems is often more complex than traditional electronic computing, and the manufacturing process is relatively new, so can its stability be guaranteed in a long time, high-load cloud computing environment? In addition, are optical computing systems likely to require additional cost inputs and special maintenance to ensure their long-term stable operation?

Response to Comment 3:

We appreciate the reviewer's insightful question regarding the long-term stability and operational costs of optical cloud computing in high-load environments. Below, we provide a detailed response to these concerns. In our proposed system, the primary addition to the communication network is the OPU clusters. The rest of the system leverages existing optical network infrastructure, minimizing fundamental changes caused by cloud computing architecture. Since optical cloud computing centers are designed to function within established optical communication networks, their integration does not introduce significant complexity beyond the OPU itself.

The active components in the OPU are subject to considerations of long-term stability, which are MZMs and PDs. Their stability is primarily governed by the long-term consistency of the MZM's U_{π} and the responsivity of the PDs under varying temperature conditions. Fluctuations in these parameters can lead to modulation inaccuracies and output drifts, introducing additional noise that ultimately degrades the precision of the computational results. To ensure consistent computational performance, the U_{π} of the modulator must remain stable over extended operation, exhibiting no more than 1 dB performance degradation after 1000 hours of continuous use. In addition, the devices are required to operate reliably across a standard indoor temperature range of 0–60 °C. Existing literature and experimental validations indicate that these optical devices can achieve long-term reliability in practical applications^{17–20}. The MZM has demonstrated stability at 85°C for over 2700 hours, confirming its long-term reliability

in high-temperature environments, with only 0.6 dB performance decrease. Meanwhile, the commercial photodetector RX10 is specified to operate within a temperature range of 0 to 70°C for more than 100 cycles, indicating its suitability for long-term computing applications with high thermal tolerance. These stability characteristics ensure the feasibility of these components for optical cloud computing. Moreover, these key components required for optical computing have already been extensively used and proven sustainable in modern optical communication systems, ensuring their feasibility for continuous cloud computing operation.

However, to maintain consistent performance in large-scale cloud computing environments, optical computing systems may require additional thermal stabilization mechanisms. While this introduces some additional power consumption, it remains a manageable factor and does not offset the overall energy efficiency advantage of optical computing. Advanced thermal control strategies, similar to those used in high-performance optical communication systems, can be adopted to ensure stable long-term operation. In the overall system, many components require temperature controllers to ensure stable operation, which contributes to the system's total energy consumption. In our experiment, the thermal stabilizer for the laser source consumes approximately 1.3 mW, while the modulator requires around 5.8 mW to maintain thermal stability. Beyond the experimentally validated data presented in this work, the AWGR also relies on a thermal stabilizer to maintain consistency. Studies have demonstrated the reliability of this structure, with the AWGR requiring approximately 600 μ W for temperature control²¹. The whole power consumption to maintain the stability of the system is shown as in Table R6.

Table R6 Estimated energy consumption of the thermal controllers

Component	Equation	numbers	Energy consumption
Programable Optical Filter	From ref. 2	1	20 mW
Laser	$P_l = P_{TEC}/\eta$	1	1.3 mW

MRM	$P_{MZM} = V_{heater}I_{heater} + V_{bias}I_{bias}$	1	5.8 mW
AWGR	From ref. ²¹	1	600 μ W
Total of thermal controllers	/	4	27.7 mW

In summary, while optical computing introduces new hardware elements, the optical components in OPU have been validated for long-term stability in the commercial optical networks. Although thermal stabilization may introduce additional power consumption, it does not diminish the broader energy efficiency benefits. We will further discuss these aspects in the revised manuscript to clarify the robustness and practicality of optical cloud computing.

Revision to Comment 3:

We have added description of the stability of our system in the *Introduction* section of the main manuscript on page 4:

“Additionally, since all the required components are commercially available, they provide assurance for the overall stability of the system.”

Additionally, we have conducted a detailed analysis of the overall system power consumption, including the additional power required to maintain system stability. The corresponding content has been added into the *Method* section:

“The power consumption of the OPU consists of two parts: the power of the transceiver, the power of the computing part and the power of the electrical control devices. They are mainly derived from the tunable optical filter, modulators, lasers, photodetectors, EDFAs and other electrical devices, as detailed in the Supplementary Note 8. To ensure the long-term operation and stability of the system, additional temperature controllers are required to maintain the functionality of the chip. This aspect has been considered in both our experiments and manuscript, with detailed records provided in the supplementary materials. Here, \$P_{laser}\$ indicates the emission power of the laser at 16 dBm, while \$P_{TEC}\$ refers to the power of the thermoelectric cooler, approximately 1.3mW. The wall-plug efficiency \$\eta \approx 0.3\$ is defined as the energy conversion

efficiency from electrical power to optical power, leading to a calculated power consumption for a single laser of approximately 137.7mW. The power consumption P_{MZM} of a MZM is calculated from the product of the bias voltage and current. As each MZM operates at a different bias voltage, the average power consumption per MZM is about 5mW. E_{MRM} represents the power consumption of MRMs, composed of the power from biasing and heaters. With a bias current of 9uA and voltage of 2V, the power consumption is dominated by the heaters. In the experiment, with a heater voltage of 2V and a current of 2.9mA, the power consumption of a single MRM is 5.8mW. P_{pd} is the energy consumption of the photodetector, estimated from $P_{pd} = RV_{bias}P_{rec}$, where $R \approx 0.65A/W$ is the responsivity of the photodetector, $V_{bias} = 2V$, and $P_{rec} = 3mW$. Thus, P_{pd} is approximately 3.9mW. EDFAs, placed before the PDs to boost the received optical power, use a broadband light source for pumping, with a central frequency λ_p close to the frequency of signal light λ_s around 1550nm. The input optical power P_{in} is 0.1mW, and the output power P_{out} is 3.1mW, with an energy conversion efficiency $\eta \approx 0.3$. Integrated multi-functional optical filters based on MZIs have been widely reported, which typically consume about 20mW and are sufficient for this system. To ensure a fair comparison, the power consumption of the electrical control module is also considered. This is mainly influenced by the DAC power, which is 40-mW. In the optical computing module, a total of 8 DACs and 6 ADCs are needed, consuming a combined power of 320.12-mW. In the communication module, 1 DAC and 1 ADC are required, occupying 20.04 mW of power. After accounting for the number of each component, the total power consumption is calculated to be 614.36-mW. However, when only considering the power consumption of the computation part, excluding the transceivers, the total energy consumption amounts to 426.92-mW. The energy efficiency can be calculated as $426.92mW/3.6TOPS = 118.6mW/TOPS$. For the long-term operation, the thermal stabilizer for the AWGR is also essential beyond the experimental setup. However, its power consumption is minimal, requiring only $600 \mu W$ ⁵⁴.”

Comment 4:

In optical cloud computing systems, with the increase in the number of parallel OPUs, time delay calibration and phase calibration can ensure synchronization between OPUs and the consistency of the optical signal phase, but does the calibration process introduce additional delays, especially when the optical path length and signal phase need to be precisely adjusted? If so, do these delays significantly affect the overall performance of the system, especially in high-speed parallel computing and real-time generation tasks?

Response to Comment 4:

We appreciate your insightful question regarding the potential additional delays introduced by time delay and phase calibration in optical cloud computing systems. Our system integrates communication and computation seamlessly, allowing synchronization overhead to be embedded within both the communication and computation signals. Each OPU's signal reception unit detects this overhead to align computational data accurately. In our experiments, we implemented frame synchronization by incorporating channel pilot synchronization sequences. Fig. R17 illustrates the frame structure of our signal synchronization sequence, which includes synchronization sequences and the payload. The payload signal is coded by the FEC sequence. The synchronization pilot aids in locating and synchronizing the signal, while the FEC sequence is utilized for error correction.

Fig. R17 The frame structure of the transmitted signal

Experiment demonstrates that signal synchronization encoding typically requires a duration of approximately 30 ns. Consequently, during high-speed synchronization processing, the calibration process introduces a delay of about 30 ns. This duration can

be neglected compared to the time required for image reception and processing, which would be hundreds of nanoseconds. Given the negligible nature of the calibration-induced delay, its impact on overall system performance is minimal. Even in scenarios demanding high-speed parallel computing and real-time generation tasks, such minor delays do not significantly affect the system's efficiency.

In summary, while time delay and phase calibration are essential for synchronizing multiple OPUs, the associated delays are minimal and do not adversely impact the performance of high-speed parallel computing or real-time tasks within optical cloud computing systems.

Revision to Comment 4:

We have added the following content regarding the calibration and delay of parallel computing in Supplementary Note 4 on Page 9:

“With the support of the aforementioned technologies, this approach enables parallel task execution. We utilize a combination of pilot-based synchronization and pre-calibration to ensure signal alignment. This process may introduce a delay exceeding 10 ns, which remains acceptable in the context of image processing. We believe that the parallel approach can significantly enhance data processing speed and improve task execution efficiency.”

Comment 5:

Minor errors on the caption of Fig. 5. “experimental result” should be “Experimental results”

Response to Comment 5:

Thank you for pointing this out. We will correct the caption in the revised manuscript.

Revision to Comment 5:

We have corrected this error in the manuscript on the caption of Fig. 5:

“Fig. 2 Experimental result of classification and generative AI tasks. a, The accuracy of MNIST handwritten digital image classification with different precision. b, Confusion matrix for MNIST handwritten digital image classification. c, Architecture of optical cloud computing system adapted for various tasks. d, Convolved waveforms from the first layer of the map edge detection task, with the red and blue lines representing the ideal and experimentally generated waveforms, respectively. e, Performance comparison between simulation and experiment with 6 different generative AI tasks (edges2handbags, edges2portrait, edges2shoes, map2edges, pix2pix-depth and segmentation). f, Image generation results for different tasks, season transfer (winter to summer; summer to winter) and semantic segmentation.”

Comment 6:

Line 42, “to consumed” should be “to consume”.

Response to Comment 6:

We appreciate you highlighting this typo. The term “to consumed” has been revised to “to consume” Thank you for your feedback.

Revision to Comment 6:

We have revised this error in Page 2, section Introduction of the Main Manuscript:
“Moreover, this growth is at the expense of a substantial energy consumption, with generative AI reported to consume \$9.5 \times 10^{12}\$ Wh of electricity¹³ in 2022”

Comment 7:

Line 295, “n the experiment” should be “in the experiment”.

Response to Comment 7:

We appreciate you highlighting this typo. The term “n the experiment” has been revised to “in the experiment” Thank you for your feedback.

Revision to Comment 7:

We have revised this error in Page 11, section Results of the Main Manuscript:
“In the experiment, a convolutional neural network was trained to handle various tasks including season transfer (winter to summer; summer to winter) and semantic segmentation, as shown in Fig. 5f.”

Once again, we sincerely thank you for your insightful feedback and constructive suggestions. We have carefully addressed your comments in the revised version of the manuscript. Your input has been instrumental in helping us refine the quality of our work, and we greatly value your time and effort in reviewing this paper.

Reference

1. Wang, L. *et al.* Enhanced-Performance Tunable Sources for Fast AWGR-Based Optical Switching Data Center Networks. *J. Light. Technol.* **42**, 8598–8605 (2024).
2. Huang, C. *et al.* A silicon photonic–electronic neural network for fibre nonlinearity compensation. *Nat. Electron.* **4**, 837–844 (2021).
3. Huang, C. *et al.* On-Chip Programmable Nonlinear Optical Signal Processor and Its Applications. *IEEE J. Sel. Top. Quantum Electron.* **27**, 1–11 (2021).
4. Zhu, H. H. *et al.* Space-efficient optical computing with an integrated chip diffractive neural network. *Nat. Commun.* **13**, 1044 (2022).
5. Shi, Y. *et al.* Nonlinear germanium-silicon photodiode for activation and monitoring in photonic neuromorphic networks. *Nat. Commun.* **13**, 6048 (2022).
6. Shrivastava, Y. & Gupta, T. K. Design of High-Speed Low Variation Static Noise Margin Ternary S-RAM Cells. *IEEE Trans. Device Mater. Reliab.* **21**, 102–110 (2021).
7. Zhang, W. *et al.* Silicon microring synapses enable photonic deep learning beyond 9-bit precision. *Optica* **9**, 579 (2022).
8. Park, Y., Hyun, J., Cho, S., Sim, B. & Lee, J. W. Any-Precision LLM: Low-Cost Deployment of Multiple, Different-Sized LLMs. Preprint at <https://doi.org/10.48550/arXiv.2402.10517> (2024).
9. Egashira, K., Vero, M., Staab, R., He, J. & Vechev, M. Exploiting LLM Quantization.
10. Song, J., Meng, C. & Ermon, S. Denoising Diffusion Implicit Models. Preprint at <https://doi.org/10.48550/arXiv.2010.02502> (2022).
11. Chu, H., Wu, W., Zang, C. & Yuan, K. QNCD: Quantization Noise Correction for Diffusion Models. in *Proceedings of the 32nd ACM International Conference on Multimedia* 10995–11003 (Association for Computing Machinery, New York, NY, USA, 2024). doi:10.1145/3664647.3681451.
12. Keysight. M8190A Arbitrary Waveform Generator. *Keysight* <https://www.keysight.com/us/en/assets/7018-02903/data-sheets/5990-7516.pdf>.
13. Shi, Y. *et al.* Avalanche photodiode with ultrahigh gain–bandwidth product of 1,033 GHz. *Nat. Photonics* **18**, 610–616 (2024).
14. Daudlin, S. *et al.* Three-dimensional photonic integration for ultra-low-energy, high-bandwidth interchip data links. *Nat. Photonics* 1–8 (2025) doi:10.1038/s41566-025-01633-0.

15. Krizhevsky, A., Sutskever, I. & Hinton, G. E. ImageNet Classification with Deep Convolutional Neural Networks. in *Advances in Neural Information Processing Systems* vol. 25 (Curran Associates, Inc., 2012).
16. Szegedy, C., Vanhoucke, V., Ioffe, S., Shlens, J. & Wojna, Z. Rethinking the Inception Architecture for Computer Vision. in *2016 IEEE Conference on Computer Vision and Pattern Recognition (CVPR)* 2818–2826 (IEEE, Las Vegas, NV, USA, 2016). doi:10.1109/CVPR.2016.308.
17. RX10AF-SpecSheet.pdf.
18. Ye, Y. *et al.* Flexible InGaAs Photodetector With High-Speed Detection and Long-Term Stability. *IEEE J. Sel. Top. Quantum Electron.* **30**, 1–8 (2024).
19. Kieninger, C. *et al.* Demonstration of long-term thermally stable silicon-organic hybrid modulators at 85 °C. *Opt. Express* **26**, 27955 (2018).
20. Eschenbaum, C. *et al.* Thermally Stable Silicon-Organic Hybrid (SOH) Mach-Zehnder Modulator for 140 GBd PAM4 transmission with Sub-1 V Drive Signals. in *2022 European Conference on Optical Communication (ECOC)* 1–4 (2022).
21. Zanetto, F. *et al.* WDM-Based Silicon Photonic Multi-Socket Interconnect Architecture With Automated Wavelength and Thermal Drift Compensation. *J. Light. Technol.* **38**, 6000–6006 (2020).

We look forward to your favorable reply. Thank you.

Yours sincerely,

Junwen Zhang

School of Information Science and Technology,

Fudan University

junwenzhang@fudan.edu.cn